# Imitation with Neural Density Models

**Kuno Kim**[1], **Akshat Jindal**[1], **Yang Song**[1], **Jiaming Song**[1], **Yanan Sui**[2], **Stefano Ermon**[1]
[1]Department of Computer Science, Stanford University
[2]NELN, School of Aerospace Engineering, Tsinghua University
Contact: khkim@cs.stanford.edu

## Abstract

We propose a new framework for Imitation Learning (IL) via density estimation of the expert's occupancy measure followed by Maximum Occupancy Entropy Reinforcement Learning (RL) using the density as a reward. Our approach maximizes a non-adversarial model-free RL objective that provably lower bounds reverse Kullback–Leibler divergence between occupancy measures of the expert and imitator. We present a practical IL algorithm, Neural Density Imitation (NDI), which obtains state-of-the-art demonstration efficiency on benchmark control tasks.

## 1 Introduction

Imitation Learning (IL) algorithms aim to learn optimal behavior by mimicking expert demonstrations. Perhaps the simplest IL method is Behavioral Cloning (BC) (Pomerleau, 1991) which ignores the dynamics of the underlying Markov Decision Process (MDP) that generated the demonstrations, and treats IL as a supervised learning problem of predicting optimal actions given states. Prior work showed that if the learned policy incurs a small BC loss, the worst case performance gap between the expert and imitator grows quadratically with the number of decision steps (Ross & Bagnell, 2010; Ross et al., 2011a). The crux of their argument is that policies that are "close" as measured by BC loss can induce disastrously different distributions over states when deployed in the environment. One family of solutions to mitigating such compounding errors is Interactive IL (Guo et al., 2014; Ross et al., 2011b, 2013), which involves running the imitator's policy and collecting corrective actions from an interactive expert. However, interactive expert queries are expensive and seldom available.

Another family of approaches (Fu et al., 2017; Ho & Ermon, 2016; Ke et al., 2020; Kim & Park, 2018; Kostrikov et al., 2020; Wang et al., 2017) that have gained much traction is to directly minimize a statistical distance between state-action distributions induced by policies of the expert and imitator, i.e the occupancy measures $\rho_{\pi_E}$ and $\rho_{\pi_\theta}$. As $\rho_{\pi_\theta}$ is an implicit distribution induced by the policy and environment[1], distribution matching with $\rho_{\pi_\theta}$ typically requires likelihood-free methods involving sampling. Sampling from $\rho_{\pi_\theta}$ entails running the imitator policy in the environment, which was not required by BC. While distribution matching IL requires additional access to an environment simulator, it has been shown to drastically improve demonstration efficiency, i.e the number of demonstrations needed to succeed at IL (Ho & Ermon, 2016). A wide suite of distribution matching IL algorithms use adversarial methods to match $\rho_{\pi_\theta}$ and $\rho_{\pi_E}$, which requires alternating between reward (discriminator) and policy (generator) updates (Fu et al., 2017; Ho & Ermon, 2016; Ke et al., 2020; Kim et al., 2019; Kostrikov et al., 2020). A key drawback to such Adversarial Imitation Learning (AIL) methods is that they inherit the instability of alternating min-max optimization (Miyato et al., 2018; Salimans et al., 2016) which is generally not guaranteed to converge (Jin et al., 2019). Furthermore, this instability is exacerbated in the IL setting where generator updates involve high-variance policy optimization and leads to sub-optimal demonstration efficiency. To alleviate this instability, (Brantley et al., 2020; Reddy et al., 2017; Wang et al., 2019) have proposed to do RL with fixed heuristic rewards. Wang et al. (2019), for example, uses a heuristic reward that estimates the

---

[1]we assume only samples can be taken from the environment dynamics and its density is unknown

support of $\rho_{\pi_E}$ which discourages the imitator from visiting out-of-support states. While having the merit of simplicity, these approaches have no guarantee of recovering the true expert policy.

In this work, we propose a new framework for IL via obtaining a density estimate $q_\phi$ of the expert's occupancy measure $\rho_{\pi_E}$ followed by Maximum Occupancy Entropy Reinforcement Learning (Max-OccEntRL) (Islam et al., 2019; Lee et al., 2019). In the MaxOccEntRL step, the density estimate $q_\phi$ is used as a *fixed reward* for RL and the occupancy entropy $\mathcal{H}(\rho_{\pi_\theta})$ is simultaneously maximized, leading to the objective $\max_\theta \mathbb{E}_{\rho_{\pi_\theta}}[\log q_\phi(s,a)] + \mathcal{H}(\rho_{\pi_\theta})$. Intuitively, our approach encourages the imitator to visit high density state-action pairs under $\rho_{\pi_E}$ while maximally exploring the state-action space. There are two main challenges to this approach. First, we require accurate density estimation of $\rho_{\pi_E}$, which is particularly challenging when the state-action space is high dimensional and the number of expert demonstrations are limited. Second, in contrast to Maximum Entropy RL (MaxEntRL), MaxOccEntRL requires maximizing the entropy of an implicit density $\rho_{\pi_\theta}$. We address the former challenge leveraging advances in density estimation (Du & Mordatch, 2018; Germain et al., 2015; Song et al., 2019). For the latter challenge, we derive a non-adversarial model-free RL objective that provably maximizes a lower bound to occupancy entropy. As a byproduct, we also obtain a model-free RL objective that lower bounds reverse Kullback-Lieber (KL) divergence between $\rho_{\pi_\theta}$ and $\rho_{\pi_E}$. The contribution of our work is introducing a novel family of distribution matching IL algorithms, named Neural Density Imitation (NDI), that (1) optimizes a principled lower bound to the additive inverse of reverse KL, thereby avoiding adversarial optimization and (2). advances state-of-the-art demonstration efficiency in IL.

## 2 Imitation Learning via density estimation

We model an agent's decision making process as a discounted infinite-horizon Markov Decision Process (MDP) $\mathcal{M} = (\mathcal{S}, \mathcal{A}, P, P_0, r, \gamma)$. Here $\mathcal{S}, \mathcal{A}$ are state-action spaces, $P : \mathcal{S} \times \mathcal{A} \to \Omega(\mathcal{S})$ is a transition dynamics where $\Omega(\mathcal{S})$ is the set of probability measures on $\mathcal{S}$, $P_0 : \mathcal{S} \to \mathbb{R}$ is an initial state distribution, $r : \mathcal{S} \times \mathcal{A} \to \mathbb{R}$ is a reward function, and $\gamma \in [0, 1)$ is a discount factor. A parameterized policy $\pi_\theta : \mathcal{S} \to \Omega(\mathcal{A})$ distills the agent's decision making rule and $\{s_t, a_t\}_{t=0}^\infty$ is the stochastic process realized by sampling an initial state from $s_0 \sim P_0(s)$ then running $\pi_\theta$ in the environment, i.e $a_t \sim \pi_\theta(\cdot|s_t), s_{t+1} \sim P(\cdot|s_t, a_t)$. We denote by $p_{\theta, t:t+k}$ the joint distribution of states $\{s_t, s_{t+1}, ..., s_{t+k}\}$, where setting $p_{\theta,t}$ recovers the marginal of $s_t$. The (unnormalized) occupancy measure of $\pi_\theta$ is defined as $\rho_{\pi_\theta}(s,a) = \sum_{t=0}^\infty \gamma^t p_{\theta,t}(s)\pi_\theta(a|s)$. Intuitively, $\rho_{\pi_\theta}(s,a)$ quantifies the frequency of visiting the state-action pair $(s, a)$ when running $\pi_\theta$ for a long time, with more emphasis on earlier states.

We denote policy performance as $J(\pi_\theta, \bar{r}) = \mathbb{E}_{\pi_\theta}[\sum_{t=0}^\infty \gamma^t \bar{r}(s_t, a_t)] = \mathbb{E}_{(s,a)\sim\rho_{\pi_\theta}}[\bar{r}(s,a)]$ where $\bar{r}$ is a (potentially) augmented reward function and $\mathbb{E}$ denotes the generalized expectation operator extended to non-normalized densities $\hat{p} : \mathcal{X} \to \mathbb{R}^+$ and functions $f : \mathcal{X} \to \mathcal{Y}$ so that $\mathbb{E}_{\hat{p}}[f(x)] = \sum_x \hat{p}(x)f(x)$. The choice of $\bar{r}$ depends on the RL framework. In standard RL, we simply have $\bar{r} = r$, while in Maximum Entropy RL (MaxEntRL) (Haarnoja et al., 2017), we have $\bar{r}(s,a) = r(s,a) - \log \pi_\theta(a|s)$. We denote the entropy of $\rho_{\pi_\theta}(s,a)$ as $\mathcal{H}(\rho_{\pi_\theta}) = \mathbb{E}_{\rho_{\pi_\theta}}[-\log \rho_{\pi_\theta}(s,a)]$ and overload notation to denote the $\gamma$-discounted causal entropy of policy $\pi_\theta$ as $\mathcal{H}(\pi_\theta) = \mathbb{E}_{\pi_\theta}[\sum_{t=0}^\infty -\gamma^t \log \pi_\theta(a_t|s_t)] = \mathbb{E}_{\rho_{\pi_\theta}}[-\log \pi_\theta(a|s)]$. Note that we use a generalized notion of entropy where the domain is extended to non-normalized densities. We can then define the Maximum Occupancy Entropy RL (MaxOccEntRL) (Islam et al., 2019; Lee et al., 2019) objective as $J(\pi_\theta, \bar{r} = r) + \mathcal{H}(\rho_{\pi_\theta})$. Note the key difference between MaxOccEntRL and MaxEntRL: entropy regularization is on the occupancy measure instead of the policy, i.e seeks state diversity instead of action diversity. We will later show in section 2.2, that a lower bound on this objective reduces to a complete model-free RL objective with an augmented reward $\bar{r}$.

Let $\pi_E, \pi_\theta$ denote an expert and imitator policy, respectively. Given only demonstrations $\mathcal{D} = \{(s,a)_i\}_{i=1}^k \sim \pi_E$ of state-action pairs sampled from the expert, Imitation Learning (IL) aims to learn a policy $\pi_\theta$ which matches the expert, i.e $\pi_\theta = \pi_E$. Formally, IL can be recast as a distribution matching problem (Ho & Ermon, 2016; Ke et al., 2020) between occupancy measures $\rho_{\pi_\theta}$ and $\rho_{\pi_E}$:

$$\text{maximize}_\theta - d(\rho_{\pi_\theta}, \rho_{\pi_E}) \tag{1}$$

where $d(\hat{p}, \hat{q})$ is a generalized statistical distance defined on the extended domain of (potentially) non-normalized probability densities $\hat{p}(x), \hat{q}(x)$ with the same normalization factor $Z > 0$, i.e $\int_x \hat{p}(x)/Z = \int_x \hat{q}(x)/Z = 1$. For $\rho_\pi$ and $\rho_{\pi_E}$, we have $Z = \frac{1}{1-\gamma}$. As we are only able to take

samples from the transition kernel and its density is unknown, $\rho_{\pi_\theta}$ is an implicit distribution[2]. Thus, optimizing Eq. 1 typically requires likelihood-free approaches leveraging samples from $\rho_{\pi_\theta}$, i.e running $\pi_\theta$ in the environment. Current state-of-the-art IL approaches use likelihood-free adversarial methods to approximately optimize Eq. 1 for various choices of $d$ such as reverse Kullback-Liebler (KL) divergence (Fu et al., 2017; Kostrikov et al., 2020) and Jensen-Shannon (JS) divergence (Ho & Ermon, 2016). However, adversarial methods are known to suffer from optimization instability which is exacerbated in the IL setting where one step in the alternating optimization involves RL.

We instead derive a non-adversarial objective for IL. In this work, we choose $d$ to be (generalized) reverse-KL divergence and leave derivations for alternate choices of $d$ to future work.

$$
\begin{aligned}
-D_{\mathrm{KL}}(\rho_{\pi_\theta}||\rho_{\pi_E}) &= \mathbb{E}_{\rho_{\pi_\theta}}[\log \rho_{\pi_E}(s,a) - \log \rho_{\pi_\theta}(s,a)] \\
&= J(\pi_\theta, \bar{r} = \log \rho_{\pi_E}) + \mathcal{H}(\rho_{\pi_\theta})
\end{aligned}
\tag{2}
$$

We see that maximizing negative reverse-KL with respect to $\pi_\theta$ is equivalent to Maximum Occupancy Entropy RL (MaxOccEntRL) with $\log \rho_{\pi_E}$ as the fixed reward. Intuitively, this objective drives $\pi_\theta$ to visit states that are most likely under $\rho_{\pi_E}$ while maximally spreading out probability mass so that if two state-action pairs are equally likely, the policy visits both. There are two main challenges associated with this approach which we address in the following sections.

1. $\log \rho_{\pi_E}$ is unknown and must be estimated from the demonstrations $\mathcal{D}$. Density estimation remains a challenging problem, especially when there are a limited number of samples and the data is high dimensional (Liu et al., 2007). Note that simply extracting the conditional $\pi(a|s)$ from an estimate of the joint $\rho_{\pi_E}(s,a)$ is an alternate way to do BC and does not resolve the compounding error problem (Ross et al., 2011a).

2. $\mathcal{H}(\rho_{\pi_\theta})$ is hard to maximize as $\rho_{\pi_\theta}$ is an implicit density. This challenge is similar to the difficulty of entropy regularizing generators (Belghazi et al., 2018; Dieng et al., 2019; Mohamed & Lakshminarayanan, 2016) for Generative Adversarial Networks (GANs) (Goodfellow et al., 2014), and most existing approaches (Dieng et al., 2019; Lee et al., 2019) use adversarial optimization.

## 2.1 Estimating the expert occupancy measure

We seek to learn a parameterized density model $q_\phi(s,a)$ of $\rho_{\pi_E}$ from samples. We consider two canonical families of density models: Autoregressive models and Energy-based models (EBMs).

**Autoregressive Models** (Germain et al., 2015; Papamakarios et al., 2017): An autoregressive model $q_\phi(x)$ for $x = (s,a)$ learns a factorized distribution of the form: $q_\phi(x) = \Pi_i q_{\phi_i}(x_i|x_{<i})$. For instance, each factor $q_{\phi_i}$ could be a mapping from $x_{<i}$ to a Gaussian density over $x_i$. When given a prior over the true dependency structure of $\{x_i\}$, this can be incorporated by refactoring the model. Autoregressive models are typically trained via Maximum Likelihood Estimation (MLE).

**Energy-based Models (EBM)** (Du & Mordatch, 2018; Song et al., 2019): Let $\mathcal{E}_\phi : \mathcal{S} \times \mathcal{A} \to \mathbb{R}$ be an energy function. An energy based model is a parameterized Boltzman distribution of the form $q_\phi(s,a) = \frac{1}{Z(\phi)} e^{-\mathcal{E}_\phi(s,a)}$, where $Z(\phi) = \int_{\mathcal{S} \times \mathcal{A}} e^{-\mathcal{E}_\phi(s,a)} ds da$ denotes the partition function. Energy-based models are desirable for high dimensional density estimation due to their expressivity, but are typically difficult to train due to the intractability of computing the partition function. However, our IL objective in Eq. 1 conveniently only requires a non-normalized density estimate as policy optimality is invariant to constant shifts in the reward. Thus, we opted to perform non-normalized density estimation with EBMs using score matching which allows us to directly learn $\mathcal{E}_\phi$ without having to estimate $Z(\phi)$.

## 2.2 Maximum Occupancy Entropy Reinforcement Learning

In general maximizing the entropy of implicit distributions is challenging due to the fact that there is no analytic form for the density function. Prior works have proposed using adversarial methods involving noise injection (Dieng et al., 2019) and fictitious play (Brown, 1951; Lee et al., 2019). We instead propose to maximize a novel lower bound to the additive inverse of an occupancy divergence which we prove is equivalent to maximizing a non-adversarial model-free RL objective. We first make clear the assumptions on the MDPs considered henceforth.

---

[2]probability models that have potentially intractable density functions, but can be sampled from to estimate expectations and gradients of expectations with respect to model parameters (Huszár, 2017).

**Assumption 1** *All considered MDPs have deterministic dynamics governed by a transition function $P : \mathcal{S} \times \mathcal{A} \to \mathcal{S}$. Furthermore, $P$ is injective with respect to $a \in \mathcal{A}$, i.e $\forall s, a, a'$ it holds that $a \neq a' \Rightarrow P(s, a) \neq P(s, a')$.*

We note that Assumption 1 holds for most continuous robotics and physics environments as they are deterministic and inverse dynamics functions $P^{-1} : \mathcal{S} \times \mathcal{S} \to \mathcal{A}$ have been successfully used in benchmark RL environments such as Mujoco (Todorov, 2014; Todorov et al., 2012) and Atari (Pathak et al., 2017). Next we introduce a crucial ingredient in deriving our occupancy entropy lower bound, which is a tractable lower bound to Mutual Information (MI) first proposed by Nguyen, Wainright, and Jordan (Nguyen et al., 2010), also known as the $f$-GAN KL (Nowozin et al., 2016) and MINE-$f$ (Belghazi et al., 2018). For random variables $X, Y$ distributed according to $p_{\theta_{xy}}(x, y), p_{\theta_x}(x), p_{\theta_y}(y)$ where $\theta = (\theta_{xy}, \theta_x, \theta_y)$, and any critic function $f : \mathcal{X} \times \mathcal{Y} \to \mathbb{R}$, it holds that $I(X; Y|\theta) \geq I_{\text{NWJ}}^f(X; Y|\theta)$ where,

$$I_{\text{NWJ}}^f(X; Y|\theta) := \mathbb{E}_{p_{\theta_{xy}}}[f(x, y)] - e^{-1}\mathbb{E}_{p_{\theta_x}}[\mathbb{E}_{p_{\theta_y}}[e^{f(x,y)}]] \tag{3}$$

This bound is tight when $f$ is chosen to be the optimal critic $f^*(x, y) = \log \frac{p_{\theta_{xy}}(x,y)}{p_{\theta_x}(x)p_{\theta_y}(y)} + 1$. We are now ready to state a lower bound to the occupancy entropy.

**Theorem 1** *Let MDP $\mathcal{M}$ satisfy assumption 1 (App. A). For any critic $f : \mathcal{S} \times \mathcal{S} \to \mathbb{R}$, it holds that*

$$\mathcal{H}(\rho_{\pi_\theta}) \geq \mathcal{H}^f(\rho_{\pi_\theta}) \tag{4}$$

*where*

$$\mathcal{H}^f(\rho_{\pi_\theta}) := \mathcal{H}(s_0) + (1 + \gamma)\mathcal{H}(\pi_\theta) + \gamma \sum_{t=0}^{\infty} \gamma^t I_{\text{NWJ}}^f(s_{t+1}; s_t|\theta) \tag{5}$$

See Appendix A.1 for the proof and a discussion of the bound tightness. Here onwards, we refer to $\mathcal{H}^f(\rho_{\pi_\theta})$ from Theorem 1 as the State-Action Entropy Lower Bound (SAELBO). The SAELBO mainly decomposes into policy entropy $\mathcal{H}(\pi_\theta)$ and Mutual Information (MI) between consecutive states $I_{\text{NWJ}}^f(s_{t+1}; s_t|\theta)$. When Assumption 1 does not hold, we may still obtain a SAELBO with only the policy entropy term, i.e $\mathcal{H}^f(\rho_{\pi_\theta}) := \mathcal{H}(\pi_\theta) \leq \mathcal{H}(\rho_{\pi_\theta})$, but this bound has more slack and is limited to discrete state-spaces. (see Appendix A for details) Since occupancy entropy maximization is also a desirable exploration strategy in sparse environments (Hazan et al., 2019; Lee et al., 2019), another interpretation of the SAELBO is as a surrogate objective for state-action level exploration. Furthermore, we posit that maximizing the SAELBO is more effective for state-action level exploration, i.e occupancy entropy maximization, than solely maximizing policy entropy. This is because, in discrete state-spaces, the SAELBO is a tighter lower bound to occupancy entropy than policy entropy, i.e $\mathcal{H}(\pi_\theta) \leq \mathcal{H}^f(\rho_{\pi_\theta}) \leq \mathcal{H}(\rho_{\pi_\theta})$, and in continuous state-spaces, where Assumption 1 holds, the SAELBO is still a lower bound while policy entropy alone is neither a lower nor upper bound to occupancy entropy. Please see Appendix C.1 for experiments that show how SAELBO maximization can improve state-action level exploration over just policy entropy maximization. Next, we show that the gradient of the SAELBO is equivalent to the gradient of a model-free RL objective.

**Theorem 2** *Let $q_\pi(a|s)$ and $\{q_t(s)\}_{t \geq 0}$ be probability densities such that $\forall s, a \in \mathcal{S} \times \mathcal{A}$ satisfy $q_\pi(a|s) = \pi_\theta(a|s)$ and $q_t(s) = p_{\theta,t}(s)$. Then for all $f : \mathcal{S} \times \mathcal{S} \to \mathbb{R}$,*

$$\nabla_\theta \mathcal{H}^f(\rho_{\pi_\theta}) = \nabla_\theta J(\pi_\theta, \bar{r} = r_\pi + r_f) \tag{6}$$

*where*

$$r_\pi(s_t, a_t) = -(1 + \gamma) \log q_\pi(a_t|s_t) \tag{7}$$

$$r_f(s_t, a_t, s_{t+1}) = \gamma f(s_t, s_{t+1}) - \frac{\gamma}{e}\mathbb{E}_{\tilde{s}_t \sim q_t, \tilde{s}_{t+1} \sim q_{t+1}}[e^{f(\tilde{s}_t, s_{t+1})} + e^{f(s_t, \tilde{s}_{t+1})}] \tag{8}$$

See Appendix A.2 for the proof. Theorem 2 shows that maximizing the SAELBO is equivalent to maximizing a discounted model-free RL objective with the reward $r_\pi + r_f$, where $r_\pi$ contributes to maximizing $\mathcal{H}(\pi_\theta)$ and $r_f$ contributes to maximizing $\sum_{t=0}^{\infty} \gamma^t I_{\text{NWJ}}^f(s_{t+1}; s_t|\theta)$. Note that evaluating $r_f$ entails estimating expectations with respect to $q_t, q_{t+1}$. This can be accomplished by rolling out multiple trajectories with the current policy and collecting the states from time-step $t, t + 1$. Alternatively, if we assume that the policy is changing slowly, we can simply take samples of states from time-step $t, t + 1$ from the replay buffer. Combining the results of Theorem 1, 2, we end the section with a lower bound on the original distribution matching objective from Eq. 1 and show that maximizing this lower bound is again, equivalent to maximizing a model-free RL objective.

---
**Algorithm 1:** Neural Density Imitation (NDI)
---
1  **Require:** Demonstrations $\mathcal{D} \sim \pi_E$, Reward weights $\lambda_\pi, \lambda_f$, Fixed critic $f$

2  ***Phase 1.*** *Density estimation*:
3  Learn $q_\phi(s, a)$ from $\mathcal{D}$ using MADE or EBMs

4  ***Phase 2.*** *MaxOccEntRL*:
5  **for** $k = 1, 2, ...$ **do**
6     Collect $(s_t, a_t, s_{t+1}, \bar{r}) \sim \pi_\theta$ and add to replay buffer $\mathcal{B}$, where
    $\bar{r} = \log q_\phi + \lambda_\pi r_\pi + \lambda_f r_f$,

$$r_\pi(s_t, a_t) = -(1 + \gamma) \log \pi_\theta(a_t | s_t)$$

$$r_f(s_t, a_t, s_{t+1}) = \gamma f(s_t, s_{t+1}) - \frac{\gamma}{e} \mathbb{E}_{\tilde{s}_t \sim \mathcal{B}_t, \tilde{s}_{t+1} \sim \mathcal{B}_{t+1}}[e^{f(s_{t+1}, \tilde{s}_t)} + e^{f(\tilde{s}_{t+1}, s_t)}]$$

    and the critic is computed by

$$f(s_{t+1}, s_t) = \log \frac{e^{-\|s_{t+1} - s_t\|_2^2}}{\mathbb{E}_{\mathcal{B}_t, \mathcal{B}_{t+1}}[e^{-\|s_{t+1} - s_t\|_2^2}]} + 1$$

    Update $\pi_\theta$ using Soft Actor-Critic (SAC) (Haarnoja et al., 2018):
7  **end**
---

**Corollary 1** *Let MDP $\mathcal{M}$ satisfy assumption 1 (App. A). For any critic $f : \mathcal{S} \times \mathcal{S} \to \mathbb{R}$, it holds that*

$$-D_{\mathrm{KL}}(\rho_{\pi_\theta} || \rho_{\pi_E}) \geq J(\pi_\theta, \bar{r} = \log \rho_{\pi_E}) + \mathcal{H}^f(\rho_{\pi_\theta}) \tag{9}$$

*Furthermore, let $r_\pi, r_f$ be defined as in Theorem 2. Then,*

$$\nabla_\theta \big( J(\pi_\theta, \bar{r} = \log \rho_{\pi_E}) + \mathcal{H}^f(\rho_{\pi_\theta}) \big) = \nabla_\theta J(\pi_\theta, \bar{r} = \log \rho_{\pi_E} + r_\pi + r_f) \tag{10}$$

In the following section we derive a practical distribution matching IL algorithm combining all the ingredients from this section.

## 3  Neural Density Imitation (NDI)

From previous section's results, we propose Neural Density Imitation (NDI) that works in two phases:

**Phase 1: Density estimation**: We leverage Autoregressive models and EBMs for density estimation of the expert's occupancy measure $\rho_{\pi_E}$ from samples. As in (Fu et al., 2017; Ho & Ermon, 2016), we take the state-action pairs in the demonstration set $\mathcal{D} = \{(s, a)_i\}_{i=1}^N \sim \pi_E$ to approximate samples from $\rho_{\pi_E}$ and fit $q_\phi$ on $\mathcal{D}$. For Autoregressive models, we use Masked Autoencoders for Density Estimation (MADE) (Germain et al., 2015) where the entire collection of conditional density models $\{q_{\phi_i}\}$ is parameterized by a single masked autoencoder network. Specifically, we use a gaussian mixture variant (Papamakarios et al., 2017) of MADE where each of the conditionals $q_{\phi_i}$ map inputs $x_{<i}$ to the mean and covariance of a gaussian mixture distribution over $x_i$. The MADE model is trained via Maximum Likelihood Estimation. With EBMs, we perform non-normalized log density estimation and thus directly parameterize the energy function $\mathcal{E}_\phi$ with neural networks since $\log q_\phi = \mathcal{E}_\phi + \log Z(\phi)$. We use Sliced Score Matching (Song et al., 2019) to train the EBM.

**Phase 2: MaxOccEntRL** After we've acquired a log density estimate $\log q_\phi$ from the previous phase, we perform RL with entropy regularization on the occupancy measure. Inspired by Corollary 1, we propose the following RL objective

$$\max_\theta J(\pi_\theta, \bar{r} = \log q_\phi + \lambda_\pi r_\pi + \lambda_f r_f) \tag{11}$$

where $\lambda_\pi, \lambda_f > 0$ are weights introduced to control the influence of the occupancy entropy regularization. In practice, Eq. 11 can be maximized using any RL algorithm by simply setting the reward function to be $\bar{r}$ from Eq. 11. In this work, we use Soft Actor-Critic (SAC) (Haarnoja et al., 2018). Note that SAC already includes a policy entropy bonus, so we do not separately include one. For our critic $f$, we *fix it* to be a normalized RBF kernel for simplicity,

$$f(s_{t+1}, s_t) = \log \frac{e^{-\|s_{t+1} - s_t\|_2^2}}{\mathbb{E}_{q_t, q_{t+1}}[e^{-\|s_{t+1} - s_t\|_2^2}]} + 1 \tag{12}$$

Table 1: Comparison between different families of distribution matching IL algorithms

| IL method | Learned Models | Relation between -divergence and optimized objective | Objective Type |
|---|---|---|---|
| Support | policy $\pi_\theta$, support estimator $f$ | Neither Upper nor Lower Bound | max |
| Adversarial | policy $\pi_\theta$, discriminator $D$ | Tight Upper Bound | min max |
| NDI (ours) | policy $\pi_\theta$, critic $f$, density $q_\phi$ | Loose Lower Bound | max max |

but future works could explore learning the critic to match the optimal critic. While simple, our choice of $f$ emulates two important properties of the optimal critic $f^*(x, y) = \log \frac{p(x|y)}{p(x)} + 1$: (1). it follows the same "form" of a log-density ratio plus a constant (2). consecutively sampled states from the joint, i.e $s_t, s_{t+1} \sim p_{\theta, t:t+1}$ have high value under our $f$ since they are likely to be close to each other under smooth dynamics, while samples from the marginals $s_t, s_{t+1} \sim q_t, q_{t+1}$ are likely to have lower value under $f$ since they can be arbitrarily different states. To estimate the expectations with respect to $q_t, q_{t+1}$ in Eq. 8, we simply take samples of previously visited states at time $t, t + 1$ from the replay buffer.

## 4 Trade-offs between Distribution Matching IL algorithms

Adversarial Imitation Learning (AIL) methods find a policy that maximizes an upperbound to the additive inverse of an $f$-divergence between the expert and imitator occupancies (Ghasemipour et al., 2019; Ke et al., 2020). For example, if the $f$-divergence is reverse KL, then for any $D : \mathcal{S} \times \mathcal{A} \to \mathbb{R}$,

$$\max_{\pi_\theta} - D_{\mathrm{KL}}(\rho_{\pi_\theta} || \rho_{\pi_E}) \leq$$
$$\max_{\pi_\theta} \, \log \left( \mathbb{E}_{\pi_E}[e^{D(s,a)}] \right) - \mathbb{E}_{\pi_\theta}[D(s,a)]$$

where the bound is tight at $D(s, a) = \log \frac{\rho_{\pi_\theta}(s,a)}{\rho_{\pi_E}(s,a)} + C$ for any constant $C$. AIL alternates between,

$$\min_{D} \, \log \left( \mathbb{E}_{\pi_E}[e^{D(s,a)}] \right) - \mathbb{E}_{\pi_\theta}[D(s,a)],$$
$$\max_{\pi_\theta} - \mathbb{E}_{\pi_\theta}[D(s,a)]$$

The discriminator update step in AIL minimizes the upper bound with respect to $D$, tightening the estimate of reverse KL, and the policy update step maximizes the tightened bound. We thus see that by using an upper bound, AIL innevitably ends up with alternating min-max optimization where policy and discriminator updates act in opposing directions. The key issue with such adversarial optimization lies not in coordinate descent itself, but in its application to a min-max objective which is widely known to gives rise to optimization instability (Salimans et al., 2016).

The key insight of NDI is to instead derive an objective that lower bounds the additive inverse of reverse KL. Recall from Eq. 9 that NDI maximizes the lower bound with the SAELBO $H^f(\rho_{\pi_\theta})$:

$$\max_{\pi_\theta} - D_{\mathrm{KL}}(\rho_{\pi_\theta} || \rho_{\pi_E}) \geq \max_{\pi_\theta} J(\pi_\theta, \bar{r} = \log \rho_{\pi_E}) + \mathcal{H}^f(\rho_{\pi_\theta})$$

Unlike the AIL upper bound, this lower bound is not tight. With critic $f$ updates, NDI alternates

$$\max_{f} \gamma \sum_{t=0}^{\infty} \gamma^t I_{\mathrm{NWJ}}^f(s_{t+1}; s_t | \theta),$$
$$\max_{\pi_\theta} J(\pi_\theta, \bar{r} = \log \rho_{\pi_E}) + (1 + \gamma)\mathcal{H}(\pi_\theta) + \gamma \sum_{t=0}^{\infty} \gamma^t I_{\mathrm{NWJ}}^f(s_{t+1}; s_t | \theta)$$

The critic update step in NDI maximizes the lower bound with respect to $f$, tightening the estimate of reverse KL, and the policy update step maximizes the tightened bound. In other words, for AIL, the policy $\pi_\theta$ and discriminator $D$ seek to push the upper bound in opposing directions while *in NDI the policy $\pi_\theta$ and critic $f$ push the lower bound in the same direction*. Unlike AIL, NDI does not perform alternating min-max but instead alternating max-max!

While NDI enjoys non-adversarial optimization, it comes at the cost of having to use a non-tight lower bound to the occupancy divergence. On the otherhand, AIL optimizes a tight upper bound at the cost of unstable alternating min-max optimization. Support matching IL algorithms also avoid min-max but their objective is neither an upper nor lower bound to the occupancy divergence. Table 1 summarizes the trade-offs between different families of algorithms for distribution matching IL.

## 5 Related Works

Prior literature on Imitation learning (IL) in the absence of an interactive expert revolves around Behavioral Cloning (BC) (Pomerleau, 1991; Wu et al., 2019), distribution matching IL (Ghasemipour et al., 2019; Ho & Ermon, 2016; Ke et al., 2020; Kim et al., 2019; Kostrikov et al., 2020; Song et al., 2018), and Inverse Reinforcement Learning (Brown et al., 2019; Fu et al., 2017; Uchibe, 2018). Many approaches in the latter category minimize statistical divergences using adversarial methods to solve a min-max optimization problem, alternating between reward (discriminator) and policy (generator) updates. ValueDICE, a more recently proposed adversarial IL approach, formulates reverse KL divergence into a completely off-policy objective thereby greatly reducing the number of environment interactions. A key issue with such Adversarial Imitation Learning (AIL) approaches is optimization instability (Jin et al., 2019; Miyato et al., 2018). Recent works have sought to avoid adversarial optimization by instead performing RL with a heuristic reward function that estimates the support of the expert occupancy measure. Random Expert Distillation (RED) (Wang et al., 2019) and Disagreement-regularized IL (Brantley et al., 2020) are two representative approaches in this family. A key limitation of these approaches is that support estimation is insufficient to recover the expert policy and thus they require an additional behavioral cloning step. Unlike AIL, we maximize a non-adversarial RL objective and unlike heuristic reward approaches, our objective provably lower bounds reverse KL between occupancy measures of the expert and imitator. Density estimation with deep neural networks is an active research area, and much progress has been made towards modeling high-dimensional structured data like images and audio. Most successful approaches parameterize a normalized probability model and estimate it with maximum likelihood, e.g., autoregressive models (Germain et al., 2015; Uria et al., 2013, 2016; van den Oord et al., 2016) and normalizing flow models (Dinh et al., 2014, 2016; Kingma & Dhariwal, 2018). Some other methods explore estimating non-normalized probability models with MCMC (Du & Mordatch, 2019; Yu et al., 2020) or training with alternative statistical divergences such as score matching (Hyvärinen, 2005; Song & Ermon, 2019; Song et al., 2019) and noise contrastive estimation (Gao et al., 2019; Gutmann & Hyvärinen, 2010). Related to MaxOccEntRL, recent works (Hazan et al., 2019; Islam et al., 2019; Lee et al., 2019) on exploration in RL have investigated state-marginal occupancy entropy maximization. To do so, (Hazan et al., 2019) requires access to a robust planning oracle, while (Lee et al., 2019) uses fictitious play, an alternative adversarial algorithm that is guaranteed to converge. Unlike these works, our approach maximizes the SAELBO which requires no planning oracle nor min-max optimization, and is trivial to implement with existing RL algorithms.

## 6 Experiments

**Environment**: Following prior work, we run experiments on benchmark Mujoco (Brockman et al., 2016; Todorov et al., 2012) tasks: Hopper (11, 3), HalfCheetah (17, 6), Walker (17, 6), Ant (111, 8), and Humanoid (376, 17), where the (observation, action) dimensions are noted parentheses.

**Pipeline**: We train expert policies using SAC (Haarnoja et al., 2018). All of our results are averaged across five random seeds where for each seed we randomly sample a trajectory from an expert, perform density estimation, and then MaxOccEntRL. Performance for each seed is averaged across 50 trajectories. For each seed we save the best imitator as measured by our augmented reward $\bar{r}$ from Eq. 11 and report its performance with respect to the ground truth reward. We don't perform sparse subsampling on the data as in (Ho & Ermon, 2016) since real world demonstration data typically aren't subsampled to such an extent and using full trajectories was sufficient to compare performance.

**Architecture**: We experiment with two variants of our method, NDI+MADE and NDI+EBM, where the only difference lies in the density model. Across all experiments, our density model $q_\phi$ is a two-layer MLP with 256 hidden units. For hyperparameters related to the MaxOccEntRL step, $\lambda_\pi = 0.2$ is fixed and for $\lambda_f$ see Section 6.3. For full details on architecture see Appendix B.

**Baselines**: We compare our method against the following baselines: (1). *Behavioral Cloning (BC)* (Pomerleau, 1991): learns a policy via direct supervised learning on $\mathcal{D}$. (2). *Random Expert Distillation (RED)* (Wang et al., 2019): estimates the support of the expert policy using a predictor

Table 2: **Task Performance** when provided with *one* demonstration. NDI (orange rows) outperforms all baselines on all tasks. See Appendix C.2 for results with varying demonstrations.

|            | HOPPER          | HALF-CHEETAH    | WALKER          | ANT             | HUMANOID        |
|------------|-----------------|-----------------|-----------------|-----------------|-----------------|
| RANDOM     | $14 \pm 8$      | $-282 \pm 80$   | $1 \pm 5$       | $-70 \pm 111$   | $123 \pm 35$    |
| BC         | $1432 \pm 382$  | $2674 \pm 633$  | $1691 \pm 1008$ | $1425 \pm 812$  | $353 \pm 171$   |
| RED        | $448 \pm 516$   | $383 \pm 819$   | $309 \pm 193$   | $910 \pm 175$   | $242 \pm 67$    |
| GAIL       | $3261 \pm 533$  | $3017 \pm 531$  | $3957 \pm 253$  | $2299 \pm 519$  | $204 \pm 67$    |
| VALUEDICE  | $2749 \pm 571$  | $3456 \pm 401$  | $3342 \pm 1514$ | $1016 \pm 313$  | $364 \pm 50$    |
| NDI+MADE   | $3288 \pm 94$   | $4119 \pm 71$   | $4518 \pm 127$  | $555 \pm 311$   | $\mathbf{6088 \pm 689}$ |
| NDI+EBM    | $\mathbf{3458 \pm 210}$ | $\mathbf{4511 \pm 569}$ | $\mathbf{5061 \pm 135}$ | $\mathbf{4293 \pm 431}$ | $5305 \pm 555$  |
| EXPERT     | $3567 \pm 4$    | $4142 \pm 132$  | $5006 \pm 472$  | $4362 \pm 827$  | $5417 \pm 2286$ |

and target network (Burda et al., 2018), followed by RL using this heuristic reward. (3). *Generative Adversarial Imitation Learning (GAIL)* (Ho & Ermon, 2016): on-policy adversarial IL method which alternates reward and policy updates. (4). *ValueDICE* (Kostrikov et al., 2020): current state-of-the-art adversarial IL method that works off-policy. See Appendix B for baseline implementation details.

## 6.1 Task Performance

Table 2 compares the ground truth reward acquired by agents trained with various IL algorithms when *one* demonstration is provided by the expert. (See Appendix C.2 for performance comparisons with varying demonstrations) *NDI+EBM achieves expert level performance on all mujoco benchmarks when provided one demonstration and outperforms all baselines on all mujoco benchmarks.* NDI+MADE achieves expert level performance on 4/5 tasks but fails on Ant. We found spurious modes in the density learned by MADE for Ant, and the RL algorithm was converging to these local maxima. We found that baselines are commonly unable to solve Humanoid with one demonstration (the most difficult task considered). RED is unable to perform well on all tasks without pretraining with BC as done in (Wang et al., 2019). For fair comparisons with methods that do not use pretraining, we also do not use pretraining for RED. See Appendix C.4 for results with a BC pretraining step added to all algorithms. GAIL and ValueDICE perform comparably with each other, both outperforming behavioral cloning. We note that these results are somewhat unsurprising given that ValueDICE (Kostrikov et al., 2020) did not claim to improve demonstration efficiency over GAIL (Ho & Ermon, 2016), but rather focused on reducing the number of environment interactions. Both methods notably under-perform the expert on Ant-v3 and Humanoid-v3 which have the largest state-action spaces. Although minimizing the number of environment interactions was not a targeted goal of this work, we found that NDI roughly requires an order of magnitude less environment interactions than GAIL. Please see Appendix C.5 for full environment sample complexity comparisons.

## 6.2 Density Evaluation

In this section, we examine the learned density model $q_\phi$ for NDI+EBM and show that it highly correlates with the true mujoco rewards which are linear functions of forward velocity. We randomly sample test states $s$ and multiple test actions $a_s$ per test state, both from a uniform distribution with boundaries at the minimum/maximum state-action values in the demonstration set. We then visualize the log marginal $\log q_\phi(s) = \log \sum_{a_s} q_\phi(s, a_s)$ projected on to two state dimensions: one corresponding to the forward velocity of the robot and the other a random selection, e.g the knee joint angle. Each point in Figure 1 corresponds to a projection of a sampled test state $s$, and the colors scale with the value of $\log q_\phi(s)$. For all environments besides Humanoid, we found that the density estimate positively correlates with velocity even on uniformly drawn state-actions which were not contained in the demonstrations. We found similar correlations for Humanoid on states in the demonstration set. Intuitively, a good density estimate should indeed have such correlations, since the true expert occupancy measure should positively correlate with forward velocity due to the expert attempting to consistently maintain high velocity.

## 6.3 Ablation studies

As intuited in Section 2.2, maximizing the SAELBO can be more effective for occupancy entropy maximization, than solely maximizing policy entropy. (see Appendix C.1 for experiments that support this) This is because in discrete state-spaces the SAELBO $\mathcal{H}^f(\rho_{\pi_\theta})$ is a tighter lower bound

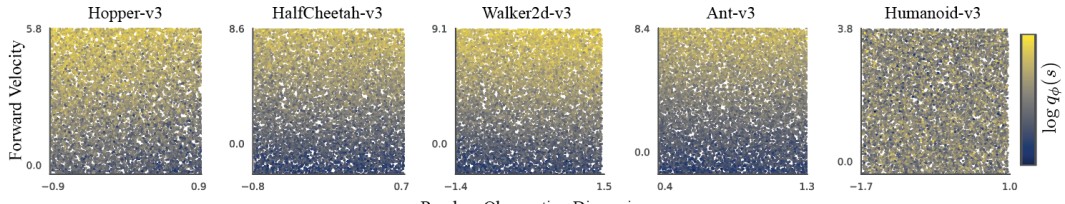

Figure 1: **Learned density visualization**. We randomly sample test states $s$ and multiple test actions $a_s$ per test state, both from a uniform distribution, then visualize the log marginal $\log q_\phi(s) = \log \sum_{a_s} q_\phi(s, a_s)$ projected onto two state dimensions: one corresponding to forward velocity and the other a random selection. Much like true reward function in Mujoco environments, we found that the log marginal positively correlates with forward velocity on 4/5 tasks.

Table 3: **Effect of varying MI reward weight** $\lambda_f$ on (1). Task performance of NDI-EBM (top row) and (2). Imitation performance of NDI-EBM (bottom row) measured as the average KL divergence between $\pi, \pi_E$ on states $s$ sampled by running $\pi$ in the true environment, i.e $\mathbb{E}_{s \sim \pi}[D_{\mathrm{KL}}(\pi(\cdot|s)||\pi_E(\cdot|s))]$, normalized by the average $D_{\mathrm{KL}}$ between the random and expert policies. $D_{\mathrm{KL}}(\pi||\pi_E)$ can be computed analytically since $\pi, \pi_E$ are conditional gaussians. Density model $q_\phi$ is trained with one demonstration. Setting $\lambda_f$ too large hurts task performance while setting it too small is suboptimal for matching the expert occupancy. A middle point of $\lambda_f = 0.005$ achieves a balance between the two metrics.

|  |  | HOPPER | HALF-CHEETAH | WALKER | ANT | HUMANOID |
|---|---|---|---|---|---|---|
| $\lambda_f = 0$ | REWARD | **3576 ± 154** | 5658 ± 698 | **5231 ± 122** | 4214 ± 444 | **5809 ± 591** |
|  | KL | 0.13 ± 0.09 | 0.35 ± 0.12 | 0.31 ± 0.08 | 0.58 ± 0.09 | 0.55 ± 0.21 |
| $\lambda_f = 0.0001$ | REWARD | 3506 ± 188 | **5697 ± 805** | 5171 ± 157 | 4158 ± 523 | 5752 ± 632 |
|  | KL | 0.15 ± 0.05 | 0.32 ± 0.15 | 0.25 ± 0.04 | 0.51 ± 0.05 | 0.41 ± 0.18 |
| $\lambda_f = 0.005$ | REWARD | 3458 ± 210 | 4511 ± 569 | 5061 ± 135 | **4293 ± 431** | 5305 ± 555 |
|  | KL | **0.11 ± 0.02** | **0.17 ± 0.09** | **0.22 ± 0.14** | **0.32 ± 0.12** | **0.12 ± 0.14** |
| $\lambda_f = 0.1$ | REWARD | 1057 ± 29 | 103 ± 59 | 2710 ± 501 | −1021 ± 21 | 142 ± 50 |
|  | KL | 0.78 ± 0.13 | 1.41 ± 0.51 | 0.41 ± 0.11 | 2.41 ± 1.41 | 0.89 ± 0.21 |
| EXPERT | REWARD | 3567 ± 4 | 4142 ± 132 | 5006 ± 472 | 4362 ± 827 | 5417 ± 2286 |

to occupancy entropy $\mathcal{H}(\rho_{\pi_\theta})$ than policy entropy $\mathcal{H}(\pi_\theta)$, i.e $\mathcal{H}(\pi_\theta) \leq \mathcal{H}^f(\rho_{\pi_\theta}) \leq \mathcal{H}(\rho_{\pi_\theta})$, and in continuous state-spaces, where Assumption 1 holds, the SAELBO is still a lower bound while policy entropy alone is neither a lower nor upper bound to occupancy entropy. As an artifact, we found that SAELBO maximization ($\lambda_f > 0$) leads to better occupancy distribution matching than sole policy entropy maximization ($\lambda_f = 0$). Table 3 shows the effect of the varying $\lambda_f$ on task (reward) and imitation performance (KL), i.e similarities between $\pi, \pi_E$ measured as $\mathbb{E}_{s \sim \pi}[D_{\mathrm{KL}}(\pi(\cdot|s)||\pi_E(\cdot|s))]$. Setting $\lambda_f$ too large ($\geq 0.1$) hurts both task and imitation performance as the MI reward $r_f$ dominates the RL objective. Setting it too small ($\leq 0.0001$), i.e only maximizing policy entropy $\mathcal{H}(\pi_\theta)$, turns out to benefit task performance, sometimes enabling the imitator to outperform the expert by concentrating most of it's trajectory probability mass to the mode of the expert's trajectory distribution. However, the boosted task performance comes at the cost of suboptimal imitation performance, e.g imitator cheetah running faster than the expert. We found that a middle point of $\lambda_f = 0.005$ simultaneously achieves expert level task performance and good imitation performance. In summary, these results show that SAELBO $\mathcal{H}^f$ maximization ($\lambda_f > 0$) improves distribution matching between $\pi, \pi_E$ over policy entropy $\mathcal{H}(\pi_\theta)$ maximization ($\lambda_f = 0$), but distribution matching may not be ideal for task performance maximization, e.g in apprenticeship learning settings. See Appendix C.1, C.3 for extended ablation studies.

# 7 Discussion and Outlook

This work's main contribution is a new principled framework for IL and an algorithm that obtains state-of-the-art demonstration efficiency. One future direction is to apply NDI to harder visual IL tasks for which AIL is known perform poorly. While the focus of this work is to improve on demonstration efficiency, another important IL performance metric is environment sample complexity. Future works could explore combining off-policy RL or model-based RL with NDI to improve on this end. Finally, there is a rich space of questions to answer regarding the effectiveness of the SAELBO reward $r_f$. We

posit that, for example, in video game environments $r_f$ may be crucial for success since state-action entropy maximization has been shown to be far more effective than policy entropy maximization (Burda et al., 2018). Furthermore, one could improve on the tightness of SAELBO by incorporating negative samples (Van Den Oord et al., 2018) and learning the critic function $f$ so that it is close to the optimal critic.

## Acknowledgements

This research was supported in part by NSF(1651565, 1522054, 1733686), ONR (N000141912145), AFOSR (FA95501910024), ARO (W911NF-21-1-0125), TRI, and Sloan Fellowship.

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
