# Imitation with Neural Density Models - Appendix

## A  Proofs

Recall the assumptions made on the MDPs.

**Assumption 1** *All considered MDPs have deterministic dynamics governed by a transition function $P : \mathcal{S} \times \mathcal{A} \to \mathcal{S}$. Furthermore, $P$ is injective with respect to $a \in \mathcal{A}$, i.e $\forall s, a, a'$ it holds that $a \neq a' \Rightarrow P(s, a) \neq P(s, a')$.*

Assumption 1 holds for most continuous robotics and physics environments as they are deterministic and inverse dynamics functions $P^{-1} : \mathcal{S} \times \mathcal{S} \to \mathcal{A}$ have been successfully used in benchmark RL environments such as Mujoco (Todorov, 2014; Todorov et al., 2012) and Atari (Pathak et al., 2017). What happens when Assumption 1 does not hold? From the proof of Theorem 1, for discrete state spaces it holds that,

$$\mathcal{H}(\rho_{\pi_\theta}(s, a)) = \mathcal{H}(\pi_\theta) + \mathcal{H}(\rho_{\pi_\theta}(s)) \geq \mathcal{H}(\pi_\theta),$$

since entropy is non-negative for discrete probability distributions, i.e $\mathcal{H}(\rho_{\pi_\theta}(s)) \geq 0$. Thus, for MDPs with discrete state-spaces, we obtain a SAELBO which is simply the policy entropy. The drawback is that this bound is less tight than the original SAELBO that includes the mutual information terms. In other words, when Assumption 1 does not hold, but the MDP state-space is discrete, one can still use NDI for non-adversarial IL which amounts to performing MaxEntRL (not MaxOccEntRL) with the learned density as the reward. As our experiments show (Table 3), optimizing only policy entropy $\mathcal{H}(\pi_\theta)$ (the row for $\lambda_f = 0$) still results in good task performance with one demonstration, but worse imitation performance (average KL). Deriving a tighter bound in the fully general space of dynamics is an interesting direction we leave for future work.

### A.1  Proof of Theorem 1

**Lemma 1** *Let $\mathcal{H} : \hat{p} \mapsto \int_{\mathcal{X}} \hat{p}(x) \log \hat{p}(x)$ denote the generalized entropy defined on the extended domain of non-normalized densities $\Delta^+ = \{\hat{p} : \mathcal{X} \to \mathbb{R}^+ \mid \exists Z > 0 \ \ s.t \ \ \int_{\mathcal{X}} \hat{p}(x)/Z = 1\}$ where $\mathbb{R}^+$ is the set of non-negative real numbers. $\mathcal{H}$ is concave.*

*Proof*

$$
\begin{aligned}
\mathcal{H}(\lambda \hat{p} + (1-\lambda)\hat{q}) &= -\sum_x \big(\lambda \hat{p}(x) + (1-\lambda)\hat{q}(x)\big) \log \big(\lambda \hat{p}(x) + (1-\lambda)\hat{q}(x)\big) \\
&\geq -\big[\sum_x \lambda \hat{p}(x) \log \hat{p}(x) + (1-\lambda)\hat{q}(x) \log \hat{q}(x)\big] \\
&= -\lambda \sum_x \hat{p}(x) \log \hat{p}(x) - (1-\lambda) \sum_x \hat{q}(x) \log \hat{q}(x) \\
&= \lambda \mathcal{H}(\hat{p}) + (1-\lambda)\mathcal{H}(\hat{q})
\end{aligned}
$$

where the inequality in the second line holds since $\forall x$, it holds that $\hat{p}(x), \hat{q}(x) \in \mathbb{R}^+$, and the map $f(u) := -u \log u$ is strictly concave on $u \in \mathbb{R}^+$; this follows from the fact that $f'(u) = -(1 + \log u)$ is strictly decreasing on $\mathbb{R}^+$. ∎

**Lemma 2** *Let MDP $\mathcal{M}$ satisfy Assumption 1. Let $\{s_t, a_t\}_{t=0}^{\infty}$ be the stochastic process realized by sampling an initial state from $s_0 \sim P_0(s)$ then running policy $\pi$ with determinsitic, injective dynamics function $P$, i.e $a_t \sim \pi(\cdot|s_t), s_{t+1} = P(s_t, a_t)$. Then $\forall t \geq 1$,*
$$\mathcal{H}(s_t|s_{t-1}) = \mathcal{H}(a_{t-1}|s_{t-1})$$

*Proof*  We expand $\mathcal{H}(s_t, a_{t-1}|s_{t-1})$ in two different ways:
$$\mathcal{H}(s_t, a_{t-1}|s_{t-1}) = \mathcal{H}(s_t|s_{t-1}, a_{t-1}) + \mathcal{H}(a_{t-1}|s_{t-1}) = 0 + \mathcal{H}(a_{t-1}|s_{t-1})$$
$$\mathcal{H}(s_t, a_{t-1}|s_{t-1}) = \mathcal{H}(a_{t-1}|s_{t-1}, s_t) + \mathcal{H}(s_t|s_{t-1}) = 0 + \mathcal{H}(s_t|s_{t-1})$$
The $\mathcal{H}(s_t|s_{t-1}, a_{t-1}), \mathcal{H}(a_{t-1}|s_{t-1}, s_t)$ terms can be zero'd out due to the determinstic, injective dynamics assumption. Thus, we conclude that $\mathcal{H}(s_t|s_{t-1}) = \mathcal{H}(a_{t-1}|s_{t-1})$. ∎

**Theorem 1** *Let MDP $\mathcal{M}$ satisfy assumption 1 (App. A). For any critic $f : \mathcal{S} \times \mathcal{S} \to \mathbb{R}$, it holds that*

$$\mathcal{H}(\rho_{\pi_\theta}) \geq \mathcal{H}^f(\rho_{\pi_\theta}) \tag{4}$$

*where*

$$\mathcal{H}^f(\rho_{\pi_\theta}) := \mathcal{H}(s_0) + (1 + \gamma)\mathcal{H}(\pi_\theta) + \gamma \sum_{t=0}^{\infty} \gamma^t I_{\text{NWJ}}^f(s_{t+1}; s_t | \theta) \tag{5}$$

*Proof*

$$
\begin{aligned}
\mathcal{H}(\rho_{\pi_\theta}(s,a)) &= -\sum_{s,a} \rho_{\pi_\theta}(s,a) \log \rho_{\pi_\theta}(s,a) \\
&= -\sum_{s,a} \rho_{\pi_\theta}(s,a) \log \frac{\rho_{\pi_\theta}(s,a)}{\rho_{\pi_\theta}(s)} - \sum_{s,a} \rho_{\pi_\theta}(s,a) \log \rho_{\pi_\theta}(s) \\
&= -\sum_{s,a} \rho_{\pi_\theta}(s,a) \log \pi_\theta(a|s) - \sum_{s} \rho_{\pi_\theta}(s) \log \rho_{\pi_\theta}(s) \\
&= \mathcal{H}(\pi_\theta) + \mathcal{H}(\rho_{\pi_\theta}(s))
\end{aligned}
$$

We now lower bound the state-marginal occupancy entropy term

$$
\begin{aligned}
\mathcal{H}(\rho_{\pi_\theta}(s)) &= \mathcal{H}(\sum_{t=0}^{\infty} \gamma^t (1-\gamma) \frac{p_{\theta,t}(s)}{1-\gamma}) \\
&\geq \sum_{t=0}^{\infty} \gamma^t \mathcal{H}(s_t) \qquad\qquad \text{Lemma 1} \qquad (13) \\
&= \left( \mathcal{H}(s_0) + \sum_{t=1}^{\infty} \gamma^t \mathcal{H}(s_t) \right) \\
&= \mathcal{H}(s_0) + \sum_{t=1}^{\infty} \gamma^t \mathcal{H}(s_t|s_{t-1}) + \sum_{t=1}^{\infty} \gamma^t I(s_t; s_{t-1})
\end{aligned}
$$

Let us consider each term separately starting with the entropy term:

$$
\begin{aligned}
\sum_{t=1}^{\infty} \gamma^t \mathcal{H}(s_t|s_{t-1}) &= \sum_{t=1}^{\infty} \gamma^t \mathcal{H}(a_{t-1}|s_{t-1}) \qquad\qquad \text{Lemma 2} \\
&= \gamma \sum_{t=0}^{\infty} \gamma^t \mathcal{H}(a_t|s_t) \\
&= \gamma \sum_{t=0}^{\infty} \gamma^t \mathbb{E}_{p(s_t,a_t)}[- \log p(a_t|s_t)] \\
&= \gamma \sum_{t=0}^{\infty} \gamma^t \mathbb{E}_{\pi_\theta}[- \log p(a_t|s_t)] \\
&= \gamma \mathbb{E}_{\pi_\theta}[- \sum_{t=0}^{\infty} \gamma^t \log \pi_\theta(a_t|s_t)] \\
&= \gamma \mathcal{H}(\pi_\theta)
\end{aligned}
$$

We now lower bound the Mutual Information (MI) term using the bound of Nguyen, Wainright, and Jordan (Nguyen et al., 2010), also known as the $f$-GAN KL (Nowozin et al., 2016) and MINE-$f$ (Belghazi et al., 2018). For random variables $X, Y$ distributed according to $p_{\theta_{xy}}(x,y), p_{\theta_x}(x), p_{\theta_y}(y)$ where $\theta = (\theta_{xy}, \theta_x, \theta_y)$, and any critic function $f(x,y)$, it holds that $I(X,Y|\theta) \geq I_{\text{NWJ}}^f(X;Y|\theta)$ where,

$$I_{\text{NWJ}}^f(X;Y) := \mathbb{E}_{p_{\theta_{xy}}}[f(x,y)] - e^{-1} \mathbb{E}_{p_{\theta_x}}[\mathbb{E}_{p_{\theta_y}}[e^{f(x,y)}]] \tag{14}$$

This bound is tight when $f$ is chosen to be the optimal critic $f^*(x, y) = \log \frac{p_{\theta_{xy}}(x,y)}{p_{\theta_x}(x)p_{\theta_y}(y)} + 1$. Applying this bound we obtain:

$$\sum_{t=1}^{\infty} \gamma^t I(s_t; s_{t-1}|\theta) \geq \sum_{t=1}^{\infty} \gamma^t I_{\text{NWJ}}^f(s_t; s_{t-1}|\theta)$$

$$= \gamma \sum_{t=0}^{\infty} \gamma^t I_{\text{NWJ}}^f(s_{t+1}; s_t|\theta)$$

Combining all the above results,

$$\mathcal{H}(\rho_{\pi_\theta}(s, a)) = \mathcal{H}(\pi_\theta) + \mathcal{H}(\rho_{\pi_\theta}(s))$$

$$= \mathcal{H}(\pi_\theta) + \mathcal{H}(s_0) + \sum_{t=1}^{\infty} \gamma^t \mathcal{H}(s_t|s_{t-1}) + \sum_{t=1}^{\infty} \gamma^t I(s_t; s_{t-1}|\theta)$$

$$\geq \mathcal{H}(s_0) + (1 + \gamma)\mathcal{H}(\pi_\theta) + \gamma \sum_{t=0}^{\infty} \gamma^t I_{\text{NWJ}}^f(s_{t+1}; s_t|\theta)$$

Setting $\mathcal{H}^f(\rho_{\pi_\theta}) := \mathcal{H}(s_0) + (1 + \gamma)\mathcal{H}(\pi_\theta) + \gamma \sum_{t=0}^{\infty} \gamma^t I_{\text{NWJ}}^f(s_{t+1}; s_t|\theta)$ concludes the proof. ∎

**Tightness of the SAELBO**: We call $\mathcal{H}^f(\rho_{\pi_\theta})$ the State-Action Entropy Lower Bound (SAELBO). There are two potential sources of slack for the SAELBO. The first source is from the application of Jensen's inequality in Eq. 13. This slack becomes smaller as $p_{\theta,t}$ converges to a stationary distribution as $t \to \infty$. The second source is the MI lowerbound $I_{\text{NWJ}}$ in Eq. 14, which can be made tight if $f$ is sufficiently flexible and chosen (or learned) to be the optimal critic.

## A.2 Proof of Theorem 2

**Theorem 2** *Let $q_\pi(a|s)$ and $\{q_t(s)\}_{t\geq 0}$ be probability densities such that $\forall s, a \in \mathcal{S} \times \mathcal{A}$ satisfy $q_\pi(a|s) = \pi_\theta(a|s)$ and $q_t(s) = p_{\theta,t}(s)$. Then for all $f : \mathcal{S} \times \mathcal{S} \rightarrow \mathbb{R}$,*

$$\nabla_\theta \mathcal{H}^f(\rho_{\pi_\theta}) = \nabla_\theta J(\pi_\theta, \bar{r} = r_\pi + r_f) \tag{6}$$

*where*

$$r_\pi(s_t, a_t) = -(1+\gamma)\log q_\pi(a_t|s_t) \tag{7}$$

$$r_f(s_t, a_t, s_{t+1}) = \gamma f(s_t, s_{t+1}) - \frac{\gamma}{e}\mathbb{E}_{\tilde{s}_t \sim q_t, \tilde{s}_{t+1} \sim q_{t+1}}[e^{f(\tilde{s}_t, s_{t+1})} + e^{f(s_t, \tilde{s}_{t+1})}] \tag{8}$$

*Proof* We take the gradient of the SAELBO $\mathcal{H}^f(\rho_{\pi_\theta})$ w.r.t $\theta$

$$\nabla_\theta \mathcal{H}^f(\rho_{\pi_\theta}) = \nabla_\theta \mathcal{H}(s_0) + \nabla_\theta(1+\gamma)\mathcal{H}(\pi_\theta) + \nabla_\theta \gamma \sum_{t=0}^{\infty} \gamma^t I_{\text{NWJ}}^f(s_{t+1}; s_t|\theta)$$

The first term vanishes, so we can consider the second and third term separately. Using the standard MaxEntRL policy gradient result (e.g Lemma A.1 of (Ho & Ermon, 2016)),

$$\nabla_\theta(1+\gamma)\mathcal{H}(\pi_\theta) = \nabla_\theta \mathbb{E}_{\pi_\theta}[-\sum_{t=0}^{\infty} \gamma^t(1+\gamma)\log q_\pi(a_t|s_t)]$$

$$= \nabla_\theta J(\pi_\theta, \bar{r} = r_\pi) \tag{15}$$

Now for the third term, we further expand the inner terms:

$$\nabla_\theta \gamma \sum_{t=0}^{\infty} \gamma^t I_{\text{NWJ}}^f(s_{t+1}; s_t|\theta)$$

$$:= \nabla_\theta \gamma \sum_{t=0}^{\infty} \gamma^t \Big( \mathbb{E}_{p_{\theta,t:t+1}(s_{t+1},s_t)}[f(s_{t+1}, s_t)] - e^{-1}\mathbb{E}_{s_{t+1} \sim p_{\theta,t+1}(s_{t+1})}[\mathbb{E}_{\tilde{s}_t \sim p_{\theta,t}(s_t)}[e^{f(s_{t+1}, \tilde{s}_t)}]] \Big)$$

$$= \nabla_\theta \gamma \sum_{t=0}^{\infty} \gamma^t \Big( \mathbb{E}_{s_0, a_0, \ldots \sim \pi_\theta}[f(s_{t+1}, s_t)] - e^{-1}\mathbb{E}_{\tilde{s}_0, \tilde{a}_0, \ldots \sim \pi_\theta}[e^{f(s_{t+1}, \tilde{s}_t)}]] \Big)$$

$$= \nabla_\theta \mathbb{E}_{s_0, a_0, \ldots \sim \pi_\theta}[\sum_{t=0}^{\infty} \gamma^{t+1} f(s_{t+1}, s_t)] - \frac{e}{\gamma}\nabla_\theta \mathbb{E}_{s_0, a_0, \ldots \sim \pi_\theta}[\sum_{t=0}^{\infty} \gamma^t \mathbb{E}_{\tilde{s}_0, \tilde{a}_0, \ldots \sim \pi_\theta}[e^{f(s_{t+1}, \tilde{s}_t)}]] \Big) \tag{16}$$

The first term is the gradient of a discounted model-free RL objective with $\bar{r}(s_t, a_t, s_{t+1}) = f(s_{t+1}, s_t)$ as the fixed reward function. The second term is not yet a model-free RL objective since the inner expectation explicitly depends on $\theta$. We further expand the second term.

$$\nabla_\theta \mathbb{E}_{s_0,a_0,\ldots\sim\pi_\theta}\left[\sum_{t=0}^\infty \gamma^t \mathbb{E}_{\tilde{s}_0,\tilde{a}_0,\ldots\sim\pi_\theta}\left[e^{f(s_{t+1},\tilde{s}_t)}\right]\right]$$

$$= \mathbb{E}_{s_0,a_0,\ldots\sim\pi_\theta}\left[\left(\sum_{t=0}^\infty \nabla_\theta \log\pi_\theta(a_t|s_t)\right)\sum_{t=0}^\infty \gamma^t \mathbb{E}_{\tilde{s}_0,\tilde{a}_0,\ldots\sim\pi_\theta}\left[e^{f(s_{t+1},\tilde{s}_t)}\right]\right]$$

$$+ \mathbb{E}_{s_0,a_0,\ldots\sim\pi_\theta}\left[\nabla_\theta \sum_{t=0}^\infty \gamma^t \mathbb{E}_{\tilde{s}_0,\tilde{a}_0,\ldots\sim\pi_\theta}\left[e^{f(s_{t+1},\tilde{s}_t)}\right]\right]$$

$$= \mathbb{E}_{s_0,a_0,\ldots\sim\pi_\theta}\left[\left(\sum_{t=0}^\infty \nabla_\theta \log\pi_\theta(a_t|s_t)\right)\sum_{t=0}^\infty \gamma^t \mathbb{E}_{\tilde{s}_0,\tilde{a}_0,\ldots\sim\pi_\theta}\left[e^{f(s_{t+1},\tilde{s}_t)}\right]\right]$$

$$+ \mathbb{E}_{s_0,a_0,\ldots\sim\pi_\theta}\left[\nabla_\theta \mathbb{E}_{\tilde{s}_0,\tilde{a}_0,\ldots\sim\pi_\theta}\left[\sum_{t=0}^\infty \gamma^t e^{f(s_{t+1},\tilde{s}_t)}\right]\right]$$

$$= \mathbb{E}_{s_0,a_0,\ldots\sim\pi_\theta}\left[\left(\sum_{t=0}^\infty \nabla_\theta \log\pi_\theta(a_t|s_t)\right)\sum_{t=0}^\infty \gamma^t \mathbb{E}_{\tilde{s}_0,\tilde{a}_0,\ldots\sim\pi_\theta}\left[e^{f(s_{t+1},\tilde{s}_t)}\right]\right]$$

$$+ \mathbb{E}_{s_0,a_0,\ldots\sim\pi_\theta}\left[\mathbb{E}_{\tilde{s}_0,\tilde{a}_0,\ldots\sim\pi_\theta}\left[\left(\sum_{t=0}^\infty \nabla_\theta \log\pi_\theta(\tilde{a}_t|\tilde{s}_t)\right)\sum_{t=0}^\infty \gamma^t e^{f(s_{t+1},\tilde{s}_t)}\right]\right]$$

$$= \mathbb{E}_{s_0,a_0,\ldots\sim\pi_\theta}\left[\left(\sum_{t=0}^\infty \nabla_\theta \log\pi_\theta(a_t|s_t)\right)\sum_{t=0}^\infty \gamma^t \mathbb{E}_{\tilde{s}_0,\tilde{a}_0,\ldots\sim\pi_\theta}\left[e^{f(s_{t+1},\tilde{s}_t)}\right]\right]$$

$$+ \mathbb{E}_{\tilde{s}_0,\tilde{a}_0,\ldots\sim\pi_\theta}\left[\mathbb{E}_{s_0,a_0,\ldots\sim\pi_\theta}\left[\left(\sum_{t=0}^\infty \nabla_\theta \log\pi_\theta(\tilde{a}_t|\tilde{s}_t)\right)\sum_{t=0}^\infty \gamma^t e^{f(s_{t+1},\tilde{s}_t)}\right]\right]$$

$$= \mathbb{E}_{s_0,a_0,\ldots\sim\pi_\theta}\left[\left(\sum_{t=0}^\infty \nabla_\theta \log\pi_\theta(a_t|s_t)\right)\sum_{t=0}^\infty \gamma^t \mathbb{E}_{\tilde{s}_0,\tilde{a}_0,\ldots\sim\pi_\theta}\left[e^{f(s_{t+1},\tilde{s}_t)}\right]\right]$$

$$+ \mathbb{E}_{\tilde{s}_0,\tilde{a}_0,\ldots\sim\pi_\theta}\left[\left(\sum_{t=0}^\infty \nabla_\theta \log\pi_\theta(\tilde{a}_t|\tilde{s}_t)\right)\sum_{t=0}^\infty \gamma^t \mathbb{E}_{s_0,a_0,\ldots\sim\pi_\theta}\left[e^{f(s_{t+1},\tilde{s}_t)}\right]\right]$$

$$= \mathbb{E}_{s_0,a_0,\ldots\sim\pi_\theta}\left[\left(\sum_{t=0}^\infty \nabla_\theta \log\pi_\theta(a_t|s_t)\right)\sum_{t=0}^\infty \gamma^t \left(\mathbb{E}_{\tilde{s}_0,\tilde{a}_0,\ldots\sim\pi_\theta}\left[e^{f(s_{t+1},\tilde{s}_t)}\right] + \mathbb{E}_{\tilde{s}_0,\tilde{a}_0,\ldots\sim\pi_\theta}\left[e^{f(\tilde{s}_{t+1},s_t)}\right]\right)\right]$$

$$= \mathbb{E}_{s_0,a_0,\ldots\sim\pi_\theta}\left[\left(\sum_{t=0}^\infty \nabla_\theta \log\pi_\theta(a_t|s_t)\right)\sum_{t=0}^\infty \gamma^t \left(\mathbb{E}_{\tilde{s}_t\sim p_{\theta,t}}\left[e^{f(s_{t+1},\tilde{s}_t)}\right] + \mathbb{E}_{\tilde{s}_{t+1}\sim p_{\theta,t+1}}\left[e^{f(\tilde{s}_{t+1},s_t)}\right]\right)\right]$$

$$= \nabla_\theta \mathbb{E}_{s_0,a_0,\ldots\sim\pi_\theta}\left[\sum_{t=0}^\infty \gamma^t \left(\mathbb{E}_{\tilde{s}_t\sim q_t}\left[e^{f(s_{t+1},\tilde{s}_t)}\right] + \mathbb{E}_{\tilde{s}_{t+1}\sim q_{t+1}}\left[e^{f(\tilde{s}_{t+1},s_t)}\right]\right)\right] \tag{17}$$

Combining the results of Eq. 16 and Eq. 17, we see that:

$$\nabla_\theta \gamma \sum_{t=0}^\infty \gamma^t I^f_{\text{NWJ}}(s_{t+1};s_t|\theta) = \nabla_\theta J(\theta, \bar{r} = r_f)$$

where, $r_f(s_t, a_t, s_{t+1}) = \gamma f(s_t, s_{t+1}) - \frac{\gamma}{e}\mathbb{E}_{\tilde{s}_t\sim q_t, \tilde{s}_{t+1}\sim q_{t+1}}[e^{f(s_{t+1},\tilde{s}_t)} + e^{f(\tilde{s}_{t+1},s_t)}]$. Finally, putting everything together with the result of Eq. 15:

$$\nabla_\theta \mathcal{H}^f(\rho_{\pi_\theta}) = \nabla_\theta J(\theta, \bar{r} = r_\pi + r_f)$$

as desired. ∎

# B  Implementation details

Here, we provide implementation details for each IL algorithm.

**NDI (Ours)**: We experiment with two variants of our method NDI+MADE and NDI+EBM, where the only difference lies in the what density estimation method was used. Across all experiments, our density model $q_\phi$ is a two-layer MLP with 256 hidden units and tanh activations. We add spectral normalization (Miyato et al., 2018) to all layers. All density models are trained with Adam (Kingma & Ba, 2014) using a learning rate of 0.0001 and batchsize 256. We train both MADE and EBM for 200 epochs. All other hyperparameters related to MADE (Germain et al., 2015) and SSM (Song et al., 2019) were taken to be the default values provided in the open-source implementations[3]. For hyperparameters related to the MaxOccEntRL step, $\lambda_\pi$ is tuned automatically in the stable-baselines implementation (Hill et al., 2018), and we set $\lambda_f = 0.005$. All RL related hyperparameters including the policy architecture are the same as those in the original SAC implementation (Haarnoja et al., 2018). We will be open-sourcing our implementation in the near future. We save the imitator with the best augmented reward as this is the model that maximizes the lower bound to reverse KL divergence.

**Behavioral Cloning** (Pomerleau, 1991): For BC, we use the stable-baselines (Hill et al., 2018) of the .pretrain() function. We parameterize the model with a two-layer MLP with 256 hidden units. Note that various forms of regularization including spectral, orthogonal, gradient penalty, $L1$, and $L2$ regularization were all found to not benefit GAIL performance and as a result the no regularization was place on the discriminator. We standardize the observations to have zero mean and unit variance (which we found drastically improves performance). We monitor the validation loss and stop training when the validation loss starts ceases to improve.

**GAIL** (Ho & Ermon, 2016): We use the stable-baselines(Hill et al., 2018) implementation of GAIL using a two-layer MLP with 256 hidden units and SAC as the RL algorithm. During training, we monitor the average discriminator reward and stop training when this reward saturates over 40 episodes.

**Random Expert Distillation** (Wang et al., 2019): We use the official implementation[4] of Random Expert Distillation (Wang et al., 2019) and explicitly set the BC pretraining flag off for all environments for the results in Table 2 and Table 5. All other hyperparameters associated with the algorithm were set to the default values that were tuned for Mujoco tasks in the original implementation. For each random seed, we sample the required number of expert trajectories and give that as the input expert trajectories to the RED algorithm. We save the model with the highest support matching reward following their protocol.

**ValueDICE** (Kostrikov et al., 2020): We use the original implementation of ValueDICE (Kostrikov et al., 2020)[5]. All hyperparameters associated with the algorithm were set to the default values that were tuned for Mujoco tasks in the original implementation. For each random seed the algorithm randomly sub-samples the required number of expert trajectories which is passed in as a flag. We conducted a hyperparameter search over the replay regularization and the number of updates performed per time step. We vary the amount of replay regularization from 0 to 0.5 and the number of updates per time step from 2 to 10 but stick with the default values as we do not find any consistent improvement in performance across environments. We save the model with the best ValueDICE loss following their protocol.

---

[3]MADE: `https://github.com/kamenbliznashki/normalizing_flows`), SSM: `https://github.com/ermongroup/ncsn`

[4]RED: `https://github.com/RuohanW/RED`

[5]ValueDICE: `https://github.com/google-research/google-research/tree/master/value_dice`

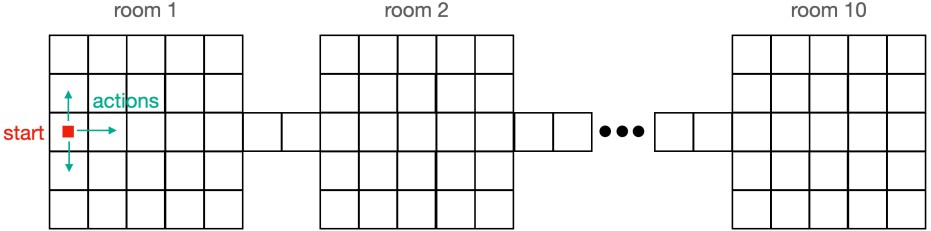

Figure 2: **Room World**. Each room is a $5 \times 5$ grid world in which the agent (red square) can move up, down, left, right (green). The agent interacts with the environment for 100 steps.

Table 4: **Effect of varying MI reward weight** $\lambda_f$ on how many rooms explored in room world

|  | NUM ROOMS EXPLORED |
| --- | --- |
| RANDOM | $1.3 \pm 0.3$ |
| $\lambda_f = 1$ (SAELBO) | $1.5 \pm 0.6$ |
| $\lambda_f = 0.1$ (SAELBO) | $3.2 \pm 0.9$ |
| $\lambda_f = 0.01$ (SAELBO) | $6.7 \pm 2.5$ |
| $\lambda_f = 0$ ($\mathcal{H}(\pi_\theta)$ ONLY) | $1.4 \pm 0.4$ |
| EXPERT | $10 \pm 0$ |

## C  Additional experiments

### C.1  Optimizing SAELBO aids exploration

In Section 2.2 we posited that maximizing the SAELBO is more effective for state-action level exploration, i.e occupancy entropy maximization, than solely maximizing policy entropy. This is because, in discrete state-spaces, the SAELBO is a tighter lower bound to occupancy entropy than policy entropy, i.e $\mathcal{H}(\pi_\theta) \leq \mathcal{H}^f(\rho_{\pi_\theta}) \leq \mathcal{H}(\rho_{\pi_\theta})$, and in continuous state-spaces, where Assumption 1 holds, the SAELBO is still a lower bound while policy entropy alone is neither a lower nor upper bound to occupancy entropy.

We designed a room world environment (see Figure 2) to illustrate how the maximizing the SAELBO $\mathcal{H}^f(\rho_{\pi_\theta})$ could enable better maximization of occupancy entropy as opposed to solely maximizing policy entropy $\mathcal{H}(\pi_\theta)$. An agent starts in a room which is linked to 10 rooms via a narrow path way. Each room is a grid world so the agent is allowed to move up, down, left, and right unless otherwise constrained by walls. The agent interacts for 100 steps. Simply encouraging the policy to be uniform random will not maximize occupancy entropy as it is unlikely for a uniform random policy to purposefully visit the 10 rooms. We compare how many rooms are visited by the agent when it maximizes only policy entropy $\mathcal{H}(\pi)$, i.e $\lambda_f = 0$, versus when it maximizes the SAELBO $\mathcal{H}^f(\rho_{\pi_\theta})$ for varying values of $\lambda_f$ with the fixed critic in Eq. 12. Note that there is no environment reward, and the agent simply tries to maximize either policy entropy or the SAELBO.

As shown in Table. 4, maximizing there is a sweet spot value of $\lambda_f = 0.01$ for which *maximizing the SAELBO allows better occupancy entropy maximization (4.8 times more rooms explored) than sole policy entropy maximization.* Please note that we do not make any strong claims that SAELBO maximization is an effective method for exploration in realistic challenging sparse reward environments. This topic is an interesting direction for future work.

Similarly, optimizing the SAELBO can be expected to improve imitation performance over solely maximizing policy entropy because the the SAELBO yields tighter lower bound to the additive inverse of reverse KL than policy entropy. Intuitively, since the SAELBO term allows better maximization of the imitator's occupancy entropy, the imitator is less likely to simply collapse all of its state-action visitation probabilities to the mode of the expert occupancy. Indeed we see in Table. 3 that including the SAELBO appropriately leads to better imitation performance.

## C.2 Task Performance with More Demonstrations

Here we present the results when using 25 expert trajectories. NDI+EBM outperforms all other methods. RED is still unable to perform well on all tasks. We found that even after hyperparameter tuning, ValueDICE and GAIL slightly underperform the expert on some tasks.

Table 5: **Task Performance** when provided with a 25 demonstrations. NDI outperforms all baselines on all tasks.

|          | HOPPER         | HALF-CHEETAH    | WALKER         | ANT             | HUMANOID         |
|----------|----------------|-----------------|----------------|-----------------|------------------|
| RANDOM   | $14 \pm 8$     | $-282 \pm 80$   | $1 \pm 5$      | $-70 \pm 111$   | $123 \pm 35$     |
| BC       | $3498 \pm 103$ | $4167 \pm 95$   | $4816 \pm 196$ | $3596 \pm 214$  | $4905 \pm 612$   |
| RED      | $2523 \pm 476$ | $-3 \pm 4$      | $1318 \pm 446$ | $1004 \pm 5$    | $2052 \pm 585$   |
| GAIL     | $3521 \pm 44$  | $3632 \pm 225$  | $4926 \pm 450$ | $3582 \pm 212$  | $259 \pm 21$     |
| VALUEDICE| $2829 \pm 685$ | $4105 \pm 134$  | $4384 \pm 620$ | $3948 \pm 350$  | $2116 \pm 1005$  |
| NDI+MADE | $3514 \pm 105$ | $4253 \pm 105$  | $4892 \pm 109$ | $1023 \pm 322$  | $6013 \pm 550$   |
| NDI+EBM  | $\mathbf{3557 \pm 109}$ | $\mathbf{5718 \pm 294}$ | $\mathbf{5210 \pm 105}$ | $\mathbf{4319 \pm 107}$ | $\mathbf{6113 \pm 210}$ |
| EXPERT   | $3567 \pm 4$   | $4142 \pm 132$  | $5006 \pm 472$ | $4362 \pm 827$  | $5417 \pm 2286$  |

## C.3 Ablation Study on each phase of NDI

Recall that NDI performs distribution matching IL in two phases: density estimation followed by MaxOccEntRL (see Section 3). Here we isolate the effect of each phase on task (Section 6.1) and imitation (Section 6.3) performance. First, we fix the MI reward weight to the optimal value $\lambda_f = 0.005$, then vary the number of demonstrations to isolate the affect of the density estimation phase on overall performance. These results are in Table 6. We see that having more demonstrations slightly improves imitation performance (KL) on most mujoco tasks. There's no clear improvement in task performance (Reward), as expert level reward is already attained with one demonstration.

Table 6: **Effect of varying the number of demonstrations** on (1). Task performance of NDI-EBM (top row) and (2). Imitation performance of NDI-EBM (bottom row) measured as the average KL divergence between $\pi, \pi_E$ on states $s$ sampled by running $\pi$ in the true environment, i.e $\mathbb{E}_{s \sim \pi}[D_{\mathrm{KL}}(\pi(\cdot|s)||\pi_E(\cdot|s))]$, normalized by the average $D_{\mathrm{KL}}$ between the random and expert policies. $D_{\mathrm{KL}}(\pi||\pi_E)$ can be computed analytically since $\pi, \pi_E$ are conditional gaussians. $\lambda$ was fixed to $\lambda_f = 0.005$. We see that having more demonstrations slightly improves imitation performance (KL) on most mujoco tasks. There's no clear improvement in task performance (Reward), as NDI is already able to achieve expert level task performance with one demonstration.

|             |        | HOPPER          | HALF-CHEETAH    | WALKER          | ANT             | HUMANOID         |
|-------------|--------|-----------------|-----------------|-----------------|-----------------|------------------|
| DEMO= 25    | REWARD | $3502 \pm 315$  | $4329 \pm 701$  | $5123 \pm 211$  | $\mathbf{4456 \pm 401}$ | $\mathbf{5513 \pm 604}$ |
|             | KL     | $\mathbf{0.10 \pm 0.01}$ | $\mathbf{0.13 \pm 0.05}$ | $0.22 \pm 0.17$ | $0.34 \pm 0.14$ | $\mathbf{0.1 \pm 0.13}$ |
| DEMO= 10    | REWARD | $3512 \pm 314$  | $4412 \pm 481$  | $\mathbf{5192 \pm 231}$ | $4311 \pm 505$  | $5381 \pm 398$   |
|             | KL     | $0.12 \pm 0.05$ | $0.15 \pm 0.06$ | $\mathbf{0.21 \pm 0.13}$ | $0.32 \pm 0.17$ | $0.11 \pm 0.15$  |
| DEMO= 4     | REWARD | $\mathbf{3552 \pm 195}$ | $\mathbf{4593 \pm 492}$ | $5153 \pm 202$  | $4342 \pm 219$  | $5310 \pm 502$   |
|             | KL     | $0.12 \pm 0.15$ | $0.18 \pm 0.07$ | $0.23 \pm 0.15$ | $0.35 \pm 0.12$ | $0.11 \pm 0.19$  |
| DEMO= 1     | REWARD | $3458 \pm 210$  | $4511 \pm 569$  | $5061 \pm 135$  | $4293 \pm 431$  | $5305 \pm 555$   |
|             | KL     | $0.11 \pm 0.02$ | $0.17 \pm 0.09$ | $0.22 \pm 0.14$ | $\mathbf{0.32 \pm 0.12}$ | $0.12 \pm 0.14$ |
| EXPERT      | REWARD | $3567 \pm 4$    | $4142 \pm 132$  | $5006 \pm 472$  | $4362 \pm 827$  | $5417 \pm 2286$  |

Next we fix the EBM density model $q_\phi(s,a)$ to be close to "oracle" by training on an ample (25) number of demonstrations then vary the strength of the MI reward $\lambda_f$ in order to isolate the effect of the MaxOccEntRL step on overall performance. These results are in Table 7. We see that, similar to results in Table 3, $\lambda_f$ mainly trades off task and imitation performance. Setting $\lambda_f$ too small drives the imitator to concentrate it's probability mass onto the modes of the expert occupancy, hence achieving good task performance at the expense of imitation performance. Setting $\lambda_f$ too large makes the entropy term dominate the objective leading to poor imitation and task performance. There's a sweet spot value of $\lambda_f = 0.005$ which balances the mode-seeking and mode-covering behavior.

Table 7: **Effect of varying MI reward weight** $\lambda_f$ on (1). Task performance of NDI-EBM (top row) and (2). Imitation performance of NDI-EBM (bottom row) measured as the average KL divergence between $\pi, \pi_E$ on states $s$ sampled by running $\pi$ in the true environment, i.e $\mathbb{E}_{s \sim \pi}[D_{\mathrm{KL}}(\pi(\cdot|s)||\pi_E(\cdot|s))]$, normalized by the average $D_{\mathrm{KL}}$ between the random and expert policies. $D_{\mathrm{KL}}(\pi||\pi_E)$ can be computed analytically since $\pi, \pi_E$ are conditional gaussians. Density model $q_\phi$ is trained with 25 demonstrations. Setting $\lambda_f$ too large hurts task performance while setting it too small is suboptimal for matching the expert occupancy. A middle point of $\lambda_f = 0.005$ achieves a balance between the two metrics.

|  |  | HOPPER | HALF-CHEETAH | WALKER | ANT | HUMANOID |
|---|---|---|---|---|---|---|
| $\lambda_f = 0$ | REWARD | **3557 ± 109** | **5718 ± 294** | 5210 ± 105 | 4319 ± 107 | **6113 ± 210** |
|  | KL | 0.14 ± 0.13 | 0.37 ± 0.1 | 0.25 ± 0.1 | 0.55 ± 0.1 | 0.57 ± 0.23 |
| $\lambda_f = 0.0001$ | REWARD | 3533 ± 163 | 5684 ± 788 | **5214 ± 201** | 4234 ± 498 | 5654 ± 541 |
|  | KL | 0.16 ± 0.07 | 0.3 ± 0.12 | 0.28 ± 0.04 | 0.56 ± 0.09 | 0.32 ± 0.12 |
| $\lambda_f = 0.005$ | REWARD | 3502 ± 315 | 4329 ± 701 | 5123 ± 211 | **4456 ± 401** | 5513 ± 604 |
|  | KL | **0.10 ± 0.01** | **0.13 ± 0.05** | **0.22 ± 0.17** | **0.34 ± 0.14** | **0.1 ± 0.13** |
| $\lambda_f = 0.1$ | REWARD | 1011 ± 124 | 221 ± 104 | 152 ± 102 | −125 ± 51 | 251 ± 103 |
|  | KL | 0.84 ± 0.16 | 1.81 ± 0.31 | 0.74 ± 0.15 | 2.92 ± 1.12 | 0.96 ± 0.43 |
| EXPERT | REWARD | 3567 ± 4 | 4142 ± 132 | 5006 ± 472 | 4362 ± 827 | 5417 ± 2286 |

## C.4 Pretraining with BC

Here we present the results obtained by pretraining all algorithms with Behavioral Cloning using 1 expert trajectory. The number of pretraining epochs was determined separately for each baseline algorithm through a simple search procedure.

For RED, we found 100 to be the optimal number of pretraining epochs and more pretraining worsens performance. For GAIL, we use 200 pretraining epochs after conducting a search from 100 to 1000 epochs. We find that the performance improves till 200 epochs and pretraining any longer worsens the performance. For ValueDICE, we use 100 pretraining epochs, determined by the same search procedure as GAIL, and found that the performance decreases when using more than 200 pretraining epochs. For NDI+MADE and NDI+EBM, we use 100 pretraining epochs.

Table 8: **Task Performance** when pretrained with BC and provided with 1 (top), 25 (bottom) expert demonstration.

|  | HOPPER | HALF-CHEETAH | WALKER | ANT | HUMANOID |
|---|---|---|---|---|---|
| RANDOM | 14 ± 8 | −282 ± 80 | 1 ± 5 | −70 ± 111 | 123 ± 35 |
| | | | 1 DEMONSTRATIONS | | |
| RED | 3390 ± 197 | 3267 ± 614 | 2260 ± 686 | 3044 ± 612 | 571 ± 191 |
| GAIL | 3500 ± 81 | 3350 ± 512 | 4175 ± 825 | 2716 ± 210 | 221 ± 48 |
| VALUEDICE | 1507 ± 308 | 3556 ± 247 | 1937 ± 912 | 1007 ± 94 | 372 ± 31 |
| NDI+MADE | 3526 ± 172 | 4152 ± 209 | 4998 ± 157 | 4014 ± 105 | 5971 ± 550 |
| NDI+EBM | **3589 ± 32** | **4622 ± 210** | **5105 ± 105** | **4412 ± 204** | 5606 ± 314 |
| | | | 25 DEMONSTRATIONS | | |
| RED | 3460 ± 153 | 3883 ± 440 | 4683 ± 994 | 4079 ± 208 | 4385 ± 1725 |
| GAIL | 3578 ± 24 | 4139 ± 275 | 4904 ± 282 | 3534 ± 346 | 281 ± 50 |
| VALUEDICE | 2124 ± 628 | 3975 ± 125 | 3939 ± 1152 | 3559 ± 134 | 101 ± 33 |
| NDI+MADE | **3533 ± 130** | 4210 ± 159 | 5010 ± 189 | 4102 ± 99 | 5103 ± 789 |
| NDI+EBM | 3489 ± 73 | **4301 ± 155** | **5102 ± 77** | **4201 ± 153** | **5501 ± 591** |
| EXPERT | 3567 ± 4 | 4142 ± 132 | 5006 ± 472 | 4362 ± 827 | 5417 ± 2286 |

We observe that the performance for RED improves drastically when pretrained with BC but is still unable to achieve expert level performance when given 1 demonstration. We observe GAIL produces better results when pretrained for 200 epochs. ValueDICE does not seem to benefit from pretraining. Pretraining also slightly improves the performance of NDI, notably in boosting the performance of NDI+MADE on Ant.

## C.5 Environment sample complexity

Although minimizing environment interactions is not a goal of this work, we show these results in Table 9 for completeness. We found that NDI roughly requires an order of magnitude less samples than GAIL which may be attributed to using a more stable non-adversarial optimization procedure. ValueDICE, an off-policy IL algorithm optimized to minimize environment sample complexity, requires roughly two orders of magnitude less interactions than NDI. We hope to see future work combine off-policy RL algorithms with NDI to further reduce environment interactions.

Table 9: **Environment Sample Complexity** computed as the mean number of environment steps needed to reach expert level performance when provided with ample (25) expert demonstrations. RED excluded as it cannot reach expert performance without BC pretraining. NDI requires less samples than GAIL but more than ValueDICE.

|  | HOPPER | HALF-CHEETAH | WALKER | ANT | HUMANOID |
|---|---|---|---|---|---|
| GAIL | $8.9M \pm 1.3M$ | $10.0M \pm 3.3M$ | $15.1M \pm 3.5M$ | $34.2M \pm 8.8M$ | $43.2M \pm 11.2M$ |
| VALUEDICE | $8.3K \pm 1.6K$ | $10.7K \pm 2.1K$ | $24.3K \pm 4.5K$ | $6.9K \pm 1.1K$ | $105K \pm 10.2K$ |
| NDI+MADE | $0.8M \pm 0.2M$ | $1.3M \pm 0.4M$ | $4.8M \pm 1.1M$ | $4.5M \pm 0.5M$ | $6.8M \pm 1.7M$ |
| NDI+EBM | $0.5M \pm 0.1M$ | $1.4M \pm 0.3M$ | $4.1M \pm 2.1M$ | $4.9M \pm 1.5M$ | $6.1M \pm 1.1M$ |