# OpenReview forum: "Imitation with Neural Density Models"
_NeurIPS.cc/2021/Conference — NeurIPS 2021 Poster_

### Official Review · Reviewer_8iqv · 2021-07-13

**Rating:** 5
**Confidence:** 3

**Summary:**

The authors propose a new imitation learning algorithm that attempts to (indirectly) minimize the KL between the imitation learner distribution and the expert distribution. The main novelty consists in the fact that the objective includes the entropy over the state-action space (as opposed to the average entropy of the action distributions given the state). In order to deal with this entropy term, the authors use a learned approximation to the state-action occupancy distribution.

**Limitations And Societal Impact:**

In the checklist, the authors say that the question of whether there are negative societal consequences of the work does not apply to this paper.  I think that the ethical questions for this paper are the same as for any other work that tries to push SOTA on imitation learning. One could for example cite the ease of construction of bots that replace humans. A short discussion of this would be nice.

**Main Review:**

The paper is original. The quality of the writing is high. The paper is mostly clear although some sections are quite dense. The proposed imitation learning method achieves the level of significance expected of a NeurIPS submission.

My main concerns are as follows:
1. The proposed method requires access to a kernel function in the state space (for computing the critic). Since we have a kernel anyway, why not use a non-parametric kernel density estimator for the state-action distribution of the expert, rather than the solutions proposed on Section 2.1 and based on neural networks? It seems strange to use the kernel for the critic but not for estimating densities.
2. What benefits does the proposed method have over MMD-GAIL [1]?.
3. You assume that the environment is deterministic. That is OK, but can you explicitly describe in the paper what breaks when the algorithm is run on a stochastic MDP?
4. One difficulty with maximising differential entropy in the state-action space is that it is not invariant to transformation of the input. Do you use raw observations provided by MuJoCo, or do you transform them somehow?
5. Because of the complexity of the proposed approach and the amount of engineering required to get good density estimates, this work is only really reproducible if the source code is released. I don't think there are enough details in the paper text itself to fully replicate the results.

Minor point:
In table 2, when you say "one demonstration" do you mean one episode or one timestep of an episode?

[1] https://www-users.cs.umn.edu/~hspark/mmd.pdf

**Time Spent Reviewing:**

5

---

> ### Author Response · Authors · 2021-08-11
> **Author Response**
>
> Thank you for your feedback and comments! We're happy you appreciate our work and would like to answer your questions.
>
> Q. Using non-parametric kernel density estimation instead of EBMs or MADE?
>
> A. We agree that this would be another interesting and natural direction to explore in the future. The density estimation step can be replaced with any estimation method of the users' choosing. We simply chose two representative classes of methods for density estimation, i.e EBMs and Autoregressive models.
>
> Q. Benefits over MMD-GAIL?
>
> A. While MMD-GAIL uses a fixed kernel to evaluate the “closeness” of the occupancies we are able to leverage the flexibility of neural networks in matching occupancies which becomes increasingly important in higher-dimensions where these occupancies are complex. It is well known that the accuracy of MMD greatly degrades with high dimensionality. (so do all density estimation methods in general but to a lesser degree than MMD)
>
> Q. Explicit description of what breaks in stochastic environments?
>
> A. As stated in line 157-159, we may still use a theoretically justified version of NDI with only the policy entropy term but only in discrete state-spaces. Practically, this means we can use the same NDI algorithm in stochastic discrete environments with the setting $\lambda_f = 0$. Further details are in line 547-560 of the Appendix. We can move more details into the main text.
>
> Q. Do we use raw Mujoco observations or transform them?
>
> A. We use the raw observations.
>
> Q. Code release?
>
> A. We will be releasing the source code.
>
> Q. Meaning of one demonstration?
>
> A. We meant one episode, i.e 1000 state-action pairs. We will clarify in the writing.
>
> Q. Add potential negative societal consequences.
>
> A. Thank you for this suggestion! We will add them to the final submission.

---

### Official Review · Reviewer_dc5s · 2021-07-15

**Rating:** 5
**Confidence:** 4

**Summary:**

The paper proposes an non-adversarial imitation learning algorithm based on density estimation of expert demonstrations. The proposed method derives a lower bound on the reverse KL divergence ($KL(\pi | \pi_{E})$), which is translated into a composite reward function consisting of 3 addictive terms. Empirical results show improved performance in Mujoco benchmarks.


**Limitations And Societal Impact:**

As discussed above, the primary limitation of the paper is that the experiments are limited in their ability to fully demonstrate the advantages and properties of the proposed method. Broadening the evaluation (e.g. different environments/tasks) would more clearly showcase the advantages of the proposed method.


**Main Review:**

Overall, the paper is well presented. The derivation for the proposed method is easy to follow and appears correct. The method is well motivated by establishing a rigorous lower bound for the reverse KL divergence.

**On non-adversarial reward**: It is difficult to assess whether the non-adversarial reward proposed in the paper improves algorithm stability, when stability is not clearly defined. In addition,  the proposed reward function appears **not fixed**, even when it’s non-adversarial. Consequently, the proposed reward function may pose similar training difficulty as the adversarial methods, since a common explanation for training difficulty stems from learning value functions under an evolving reward. The experiments seem to suggest that $\lambda_f$  needs to be small (i.e. reduce how much the reward function changes) to obtain good performances, while the best policy performance is obtained with $\lambda_f = 0$. More generally, could the authors further discuss how much the dynamic terms actually contribute to the reward function (including the auto-tuned value for $\lambda_\pi$), and how they affect policy learning?

**Efficiency wrt demonstrations**: This claim is difficult to support with the current experiments. Based on the existing conventions, 1 full trajectory contains 1000 state-action pairs, while GAIL and some previous methods use 4 sub-sampled trajectories of 50 state-actions each (200 state-actions in total, except for the Humanoid task). It is therefore unclear that the proposed method is more sample efficient wrt demonstrations. Could authors further discuss the differences between the two settings and their implications for the proposed method?

**Practical algorithm**: The reward in the form of log(x) is typically challenging to optimize, since the reward has a very large range. In addition, the proposed method has three terms with possibly different ranges. Could the authors clarify (if any) modification in the actual implementation for “normalizing” the reward?

To my knowledge, SAC is already an off-policy algorithm. The limitation “Future works could explore combining off-policy RL or model-based RL with NDI to improve on (environment sample complexity)” is thus confusing to me. Please clarify this point.

**Time Spent Reviewing:**

6

---

> ### Author Response · Authors · 2021-08-11
> **Author response**
>
> Thank you very much for committing your time to reviewing our paper and for the thoughtful questions and feedback. We'd like to answer all of your questions, resolve any potential misunderstandings, and address your feedback.
>
> Q. Experiments are limited in fully demonstrating the advantages and properties of the proposed method. Broadening the evaluation (e.g. different environments/tasks) would more clearly showcase the advantages of the proposed method.
>
> A. Thank you for this feedback. We'd like to emphasize that NDI achieves state-of-the-art performance on the Mujoco benchmark dataset which is the single dataset of evaluation in many keystone imitation learning papers, e.g GAIL, MAIL, ValueDICE, DAC. Furthermore, we include the Humanoid task and achieve successful imitation with 1 demonstration while in many prior works Humanoid is excluded due to its challenging nature. Thus, we believe we're able to clearly demonstrate the advantages of NDI with respect to prior works. If there are additional datasets the reviewer would like to see experiments on please let us know during the discussion phase.
>
> Q. Evaluation of stability? Lingering training instability due to an evolving reward function?
>
> A. We can include the full learning curves to show improved learning stability if the reviewer would like to see them. While we agree that training instability can be introduced by evolving rewards, we also believe that the “evolving nature” itself not the primary cause of instability in adversarial imitation learning methods that need to be resolved. Rather the min-max objective combined with coordinate descent is the main problem as discussed in line 220-225 of the submission and our work seeks to resolve this piece of the problem. There are several non-adversarial RL objectives (as is our method) with an evolving reward that work well in practice. For example, MaxEntRL also uses an evolving reward (since $\log \pi$ is changing) and the SAC paper shows the evolving reward term is in fact critical for good performance. Curiosity driven RL including Random Network Distillation (Burda et all 2020), ICM (Pathak 2017), and AWML (Kim et al. 2020) all optimize evolving rewards.
>
> Q. $\lambda_f$ needs to be small for best policy performance. How much do dynamic terms contribute to policy learning?
>
> A. Best task performance (reward) does not necessarily mean good imitation learning performance. If the demonstrator is sub-optimal than a good “imitator” would also be suboptimal. While $\lambda_f = 0$ maximizes task performance (reward), it results in the imitator outperforming the demonstrator which results in poor imitation performance (KL) as shown in Table 3. Thus, the mutual information term $\lambda_f$ mainly contributes to “imitation” performance, i.e KL between the imitator and expert’s policy, and finding a sweet-spot value leads to optimal balance between reward and imitation performance. The policy entropy term $\lambda_\pi$ is the main knob controlling the degree of “exploration” in RL and is thus critical for good performance of SAC as shown in their paper. Another perspective is that $\lambda_\pi$ makes the imitator's trajectory distribution more diffuse, preventing placement of all probability mass on the modes of the expert trajectory distribution. We can run an additional ablation on $\lambda_\pi$ if the reviewer would like to see this data.
>
> Q. Some baselines, e.g GAIL, use subsampled state-action pairs while NDI uses all 1000.
>
> A. NDI and all baselines (including GAIL) were given exactly the same amount of data: 1000 state-action pairs per trajectory without sub-sampling in order to have a fair comparison of demonstration efficiency. The subsampling setting most likely yields more information from the same number of state-action pairs. For example, using 10 subsampled trajectories with 100 state-actions from each (yielding a total of 1000 state-action pairs) is most likely a more informative demonstration dataset than one un-subsampled trajectory with 1000 state-action pairs. This is because a lot of the neighboring time state-action pairs are essentially the same and provide less meaningful information than state-action pairs from diverse trajectories which give information on how to act in vastly different states. Thus given the same amount of state-action pairs our non-subsampled setting is likely more challenging. We chose the non-subsampled setting as we found that the subsampled setting is rather a bit artificial in that it is difficult to imagine realistic scenarios in which you’ve already collected a full trajectory but only have access to a sub-sampled portion of it.
>
> Q. Any reward normalization?
>
> A. We have borrowed some standard reward normalization techniques from RL and reward clipping techniques. We will add these details to the final submission.
>
> Q. Clarification on “combining off-policy RL” limitation
>
> A. Thank you for catching this. It was a typo, we meant “more advanced off-policy RL methods” since there have been some advances in off-policy RL since SAC.

---

### Official Review · Reviewer_wy8U · 2021-07-17

**Rating:** 7
**Confidence:** 2

**Summary:**

This paper presents an imitation learning approach based on density estimation of the state-action pairs. The underlying idea is to take expert demonstrations and find a distribution over the inherent state occupancy and then follow-up with a reinforcement learning step. As a result, the learner visits the high density state action pairs while also exploring new states during training. The paper makes a number of interesting contributions, among a new cost function for distribution matching IL and a proof showing that it is equivalent to maximizing a non-adversarial model-free RL objective. In contrast to other methods, like AIL, no adversarial optimization is needed; which typically would have resulted in instabilities. Instead the new objective pushes the lower bound in a single direction. The paper also features a list of tradeoffs between different distribution matching IL methods. Experiments on Mujoco tasks show state of the art performance.

**Limitations And Societal Impact:**

The authors talk about limitations and provide a table of tradeoffs when compared to other methods.

**Main Review:**

I generally enjoyed this paper. It starts with a good background on modern IL methods and their relationships and goes on to pinpoint the challenges to be addressed. Obviously, the idea of distribution matching IL algorithms is not new. However, this paper goes about it in a rigorous and well-motivated way. On one hand, it shows how to maximize an implicit density and how to avoid the instabilities that afflict other methods. Generally, a bunch of small but non-trivial tricks are combined to make this algorithm work. The density estimation step itself is performed via existing methods, e.g., EBMs. While not necessarily completely original, the strength of the paper is in a deep analysis and careful and rigorous derivation of an improved distribution matching IL algorithm.

The quality of writing is very high. Overall, it's always clear what the authors are trying to achieve. However, at times parsing the paper can become difficult due to the mathematical notation and dense mixtures of equations and text. I would recommend making another pass for readability and accessibility. In particular, some long sentences can be cut down into pieces. As for the quality of the presented ideas, I think this is a good paper with important theoretical and practical contributions to imitation learning. As mentioned before, it provides a deep dive into the intricacies of modern distribution matching IL methods. I am slightly confused however why the authors chose to show two variants, NDI+MADE / NDI+EBM? Is the underlying idea to show that the method is extensible?

The presented results in the mujoco task outperform the other SotA methods and clearly show the advantage of the approach. One important piece of information that I couldn't find is "how many RL iterations have been performed for the experiments?". Maybe the authors can comment on that.  Overall, this is a good paper with interesting insights, a technically sound derivation and great results in simulation.

BTW, the paper mentions code but I could not find any code for verification in the supplemental material.

Edit after rebuttal: Thanks to the authors for the comprehensive and satisfying answers. I am maintaining my score. This is a good paper and I argue for acceptance.

**Time Spent Reviewing:**

7

---

> ### Author Response · Authors · 2021-08-11
> **Author Response**
>
> Thank you for your time and effort in reviewing our paper. We are happy that you appreciate our work! We'd like to answer your questions.
>
> Q. Take another pass for readability
>
> A. Thank you for the suggestion, we will do so.
>
> Q. Why show two variants NDI + MADE / NDI + EBM?
>
> A. We sought to show that our framework is compatible with two different representative methods of density estimation and give the message that users’ can replace the density estimation step with other methods of their choosing.
>
> Q. How many RL iterations have been performed for experiments?
>
> A. The number of RL update steps is roughly equal to the number of environment interactions since one update is performed per environment interaction. Environment sample complexity can be found in Appendix C.5.
>
> Q. Location of code?
>
> A. We will be releasing the code shortly after publication.

---

### Official Review · Reviewer_VQpP · 2021-07-20

**Rating:** 7
**Confidence:** 3

**Summary:**

Online imitation learning algorithm based on neural density estimation is proposed. The objective of this paper is to maximize $-D_{KL}$ in Eq. (2), and the authors consider maximizing the lower bound of Eq. (2) in Corollary 1 as a surrogate learning objective. The aforementioned neural density model is used to estimate the expert state-action visitation distribution (or occupancy measure) which works as a component of a reward function (the log of the neural density model) during the RL phase, together with the entropy regularization ($r_\pi$) and a critic-based reward ($r_f$). While Adversarial Imitation Learning requires the alternate optimization for policy and reward estimators, the proposed idea does not require the reward learning phase (similar to RED, Disagreement regularized IL and SQIL), which may stabilize the learning procedure. While the proposed idea is straightforward to be understood, the empirical results show that NDI outperforms its baselines especially when the number of expert demonstrations is small. The ablation studies on learned density and varying MI reward weight are also interesting to follow.

**Limitations And Societal Impact:**

No societal issue provided.

**Main Review:**

I think the submission is clearly written, and theoretical challenges and how to address them are well-described. A wide range of baselines are considered and compared with the proposed idea, and the experiment supports the authors’ claim. I couldn’t find any major flaw in this work overall, and I had minor questions below:
- While I agree that using an unnormalized density estimate is okay in theory, I wonder if it doesn’t affect the algorithmic stability since the reward may be extremely large or small.
- I read through Islan et al., and Lee et al. but couldn’t find the term “Maximum Occupancy Entropy RL”. Is this named by the author?
- One question I had while evaluating this submission was if the discounted sum over causal entropies and the entropy of occupancy measure are different. I concluded they are different based on the theoretical result in GAIL’s Lemma 3.2, and the difference is related to the entropy of the state-visitation occupancy measure. I would like authors to more emphasize the difference between the two.


**Time Spent Reviewing:**

5

---

> ### Author Response · Authors · 2021-08-11
> **Author Response**
>
> Thank you for your detailed comments and feedback! We are happy that you appreciate our work and would like to answer your questions.
>
> Q. Effect of unnormalized density estimates as reward
>
> A. We used standard reward normalization techniques in RL as well as reward clipping. We will add these details in the final submission.
>
> Q. Did we name the term Maximum Occupancy Entropy RL?
>
> A. To our knowledge, yes we have first used this naming.
>
> Q. More emphasis on difference between discounted causal entropy and occupancy entropy.
>
> A. Thank you for this suggestion, we will revise to make the distinction more clear. Indeed the two can be very different quantities. A uniform random policy has maximum discounted causal entropy, i.e policy entropy, however acting uniformly random in environments are unlikely to get you to visit diverse states, e.g in maze environments. Hence the same policy would have low occupancy entropy.

---

### Decision · Program_Chairs · 2021-09-27

**Decision:**

Accept (Poster)

**Comment:**

This paper proposes a new framework for imitation learning via obtaining a density estimation of the expert’s occupancy measure, followed by Maximum Occupancy Entropy Reinforcement Learning. The intuitive idea is to encourage the imitator to visit high density state-action pairs under the expert’s occupancy measure while maximally exploring the state-action space.

In their initial reviews, the reviewers find the paper well-written. The proposed method is based on in-depth analysis of the problem and it has strong performance in comparison with a wide range of baselines. On the other hand, they also think that the claim on data efficiency not is well supported and lack some necessary discussions (evolving reward and stability of learning, use of non-parametric kernel density, advantages over MMG-GAIL, etc)

During discussions, two reviewers stood by their votes for accept. Two other reviewers felt that their concerns were not satisfactorily addressed by authors’ responses. One of them even lowered his/her score.